# NLRP11 attenuates Toll-like receptor signalling by targeting TRAF6 for degradation via the ubiquitin ligase RNF19A

Chenglei Wu[1], Zexiong Su[2], Meng Lin[2], Jiayu Ou[1], Wei Zhao[1], Jun Cui[2,3] & Rong-Fu Wang[4,5,6]

The adaptor protein TRAF6 has a central function in Toll-like receptor (TLR) signalling, yet the molecular mechanisms controlling its activity and stability are unclear. Here we show that NLRP11, a primate specific gene, inhibits TLR signalling by targeting TRAF6 for degradation. NLRP11 recruits the ubiquitin ligase RNF19A to catalyze K48-linked ubiquitination of TRAF6 at multiple sites, thereby leading to the degradation of TRAF6. Furthermore, deficiency in either NLRP11 or RNF19A abrogates K48-linked ubiquitination and degradation of TRAF6, which promotes activation of NF-κB and MAPK signalling and increases the production of proinflammatory cytokines. Therefore, our findings identify NLRP11 as a conserved negative regulator of TLR signalling in primate cells and reveal a mechanism by which the NLRP11-RNF19A axis targets TRAF6 for degradation.

[1] Zhongshan School of Medicine, Sun Yat-sen University, Guangzhou 510080, People's Republic of China. [2] Key Laboratory of Gene Engineering of the Ministry of Education, State Key Laboratory of Biocontrol, School of Life Sciences, Sun Yat-sen University, Guangzhou 510006, People's Republic of China. [3] Collaborative Innovation Center of Cancer Medicine, Sun Yat-sen University, Guangzhou, Guangdong 510080, People's Republic of China. [4] Center for Inflammation and Epigenetics, Houston Methodist Research Institute, Houston, TX 77030, USA. [5] Department of Microbiology and Immunology, Weill Cornell Medicine, Cornell University, New York, NY 10065, USA. [6] Institute of Biosciences and Technology, College of Medicine, Texas A & M University, Houston, TX 77030, USA. Correspondence and requests for materials should be addressed to J.C. (email: cuij5@mail.sysu.edu.cn) or to R.-F.W. (email: rwang3@houstonmethodist.org)

Recognition of microbial pathogens is crucial for the initiation of innate immune responses and is mediated by germline-encoded pattern-recognition receptors (PRR) that detect conserved features of invading microorganisms, termed pathogen associated molecular patterns (PAMP)[1–5]. Subfamilies of PRRs include Toll-like receptors (TLR), RIG-I-like receptors (RLR), NOD-like receptors (NLR), and several DNA sensors.

Upon PAMP recognition, PRRs induce a series of signalling cascades that induce proinflammatory cytokine and type I interferon (IFN) production, which coordinately result in protection from invading pathogens.

Upon detection of PAMPs, most TLRs associate with MyD88, which in turn recruits proteins of the IRAK family[4,6,7]. IRAK4 is initially recruited to the TLR complex and subsequently

**Fig. 1** Expression, protein stability and intracellular localization of NLRP11. **a** *NLRP11* mRNA in different human tissues was analyzed by real-time PCR. **b** Real-time PCR analysis of NLRP11 expression in PBMCs, T cells, B cells, and monocytes. **c, d** THP-1 cells (a human monocytic leukemia cell line) **c** and PBMCs **d** were stimulated with LPS (100 ng/ml, a TLR4 ligand) at the indicated time points, then the transcription of NLRP11 was anyalyzed by real-time PCR. **e, f** Immunoblot analysis of NLRP11 expression in THP-1 cells treated with LPS (100 ng/ml) **e** or Pam3CSK4 (100 ng/ml, a TLR2 ligand) **f** for the indicated periods. **g** PBMCs were exposed to Pam3CSK4 (100 ng/ml), poly(I:C) (10 μg/ml, a TLR3 ligand), LPS, or PBS for 8 h, and the cell lysates were immunoblotted with an anti-NLRP11 or β-actin antibody. **h** Immunoblot analysis of NLRP11 expression in THP-1 cells pretreated with BAY11-7082 (10 μM, an inhibitor of IκBα) for 1 h, then stimulated with LPS for 6 h. **i** Wild type (WT) and MyD88$^{-/-}$ THP-1 cells were stimulated with LPS and used for measuring its induction by real-time PCR. **j** 293T cells transfected with FLAG-tagged NLRP11 were either untreated or pretreated with MG132 for 6 h, then the cells were incubated with cycloheximide (CHX; protein synthesis inhibitor) and were immunoblotted with the indicated antibodies. **k** THP-1 cells were untreated or pretreated with MG132 for 6 h, then incubated with CHX for the indicated periods and were immunoblotted with indicated antibodies. **l** The THP-1 cells were stimulated or unstimulated with LPS for 8 h, and cytoplasmic and nuclear extracts of THP-1 cells were treated or untreated with 50 nM LMB for 4 h were analyzed by immunoblot with antibodies to NLRP11, tubulin (cytoplasmic fraction), and lamin (nuclear fraction). *$P < 0.05$, **$P < 0.01$ using Student's t-test. Data **a–d** and **i** are presented as mean ± SEM of combined from three independent experiments. Experiments **e–h**, **j–l** are representatives of three independent experiments with similar results. Data in **j** and **k** are presented as means ± SEM of combined from three independent experiments

phosphorylates and activates IRAK1 and IRAK2[8,9]. Activated IRAKs promote oligomerization of TRAF6, which induces its ubiquitin ligase activity. TRAF6 then forms a complex with the E2 ubiquitin-conjugating enzyme complex Ubc13-Uev1A to catalyze Lys (K) 63-linked poly-ubiquitination on itself or other substrates[10]. Ubiquitinated TRAF6 then activates transforming growth factor β-activated protein kinase 1 (TAK1) in a ubiquitin-dependent manner; this action results in the activation of NF-κB and MAPKs and subsequent production of cytokines and chemokines[11,12]. Thus, tight regulation of TRAF6 activity is necessary for activation of an appropriate immune response. Strong evidence exists that K63-linked poly-ubiquitination is important for the activation of TRAF6, and several deubiquitinases can attenuate TRAF6 activity by directly removing the ubiquitin chains from TRAF6, including A20, CYLD, USP2a, USP4, and USP20[13–19]. UBE2O, a putative E2 ubiquitin-conjugating (UBC) enzyme, inhibits TRAF6-mediated NF-κB activation by disrupting the interaction between MyD88 and TRAF6[20]. The kinase MST4 has been shown to directly phosphorylate TRAF6 to inhibit TLR4 signalling[21]. In addition, TRIM38 has been reported to target TRAF6 for K48-linked poly-ubiquitination and proteasomal degradation in mouse cells[22]. Nonetheless, whether additional regulators are responsible for K48-linked ubiquitination of TRAF6 for regulation of its stability is unknown.

NLR proteins are a family of intracellular PRRs that share a typical nucleotide-binding and oligomerization domain (NOD) and a leucine-rich repeat (LRR) region, but a variable N-terminal effector domain. Several intensively studied NLRs, including NLRP1, NLRP3, NLRC4, Nod1, and Nod2, are well known to form the inflammasome complex or trigger an innate immune response[23–26]. Several NLRs have also been identified as negative regulators of TLR signalling[24,27,28]. NLRC3 is reported to suppress TLR-dependent NF-κB activation by deubiquitinating TRAF6[29], and NLRX1 to inhibit TLR signalling by interacting with TRAF6 and IKKβ[30,31]. NLRC5, an important transactivator of MHC class I, also has a central function in the negative regulation of NF-κB and type I interferon pathways via associations with IKKα/β and RIG-1/MDA5, respectively[32–34]. Here we report that NLRP11, a human only NLR, is a negative regulator of TLR signalling by targeting TRAF6. As an NF-κB-inducible gene, NLRP11 recruits the ubiquitin ligase RNF19A to catalyze K48-linked ubiquitination of TRAF6, thereby leading to destabilization of the TRAF6 protein to negatively feedback and terminate TLR signalling. Our results identify the NLRP11-RNF19A axis as a signalling complex for TRAF6 degradation to prevent a dysregulated inflammatory response.

## Results

**Expression and intracellular localization of NLRP11.** As a member of the NLR protein family, NLRP11 contains an N-terminal PYD domain, a central NOD domain, and a C-terminal LRR domain, but the role of NLRP11 in the regulation of inflammatory responses has not yet been elucidated. NLRP11, a primate specific gene, has been reported to be expressed in macaque ovary and THP-1 cells (a human monocytic leukemia cell line)[35–37]. Here we examined the expression of NLRP11 in various human tissues by real-time PCR analysis. The results confirmed that NLRP11 is highly expressed in the testis, ovary, and lung, and weakly expressed in other tissues (Fig. 1a). To characterize NLRP11 expression in immune cells, we analyzed NLRP11 transcript abundance in T cells, B cells, and monocytes isolated from peripheral blood mononuclear cells (PBMCs). We found that monocytes showed the highest expression levels of NLRP11, suggesting that NLRP11 may play a significant role in monocytes (Fig. 1b, Supplementary Fig. 1a). We further examined

NLRP11 expression in THP-1 cells, THP-1-derived macrophages, and peripheral blood mononuclear cells (PBMCs) under stimulation with LPS (a ligand of TLR4). We found that LPS treatment markedly induced NLRP11 in these cells (Fig. 1c, d, Supplementary Fig. 1b). In line with this finding, NLRP11 protein in THP-1 cells was also upregulated in response to LPS treatment (Fig. 1e). We next explored to determine whether other TLR ligands have a similar effect. Indeed, we found that in addition to LPS, Pam3CSK4 (a TLR1/TLR2 ligand) can enhance NLRP11 expression in THP-1 cells and PBMCs, whereas expression of the NLRP11 protein was weakly changed after poly(I:C) treatment (a TLR3 ligand; Fig. 1f, g). Moreover, the expression of NLRP11 in dendritic cells (DCs) is relatively low compared to PBMCs and could not be induced by LPS or Pam3CSK4 (Supplementary Fig. 1c). Interestingly, we further found that other non-TLR NF-κB inducers, such as TNF or PMA, can induce the expression of NLRP11 in THP-1 cells, suggesting that NLRP11 is a common NF-κB inducible gene (Supplementary Fig. 1d). Since TLR1/TLR2 and TLR4 mainly utilize the same adaptor MyD88 to strongly activate NF-κB signalling, whereas TLR3 mainly uses TRIF to induce type I IFN activation and weakly induce NF-κB activation, we thus speculated that the expression of NLRP11 is regulated in a NF-κB dependent manner. To test this prediction, we first assessed NLRP11 expression in THP-1 cells pretreated with BAY11-7082 (an inhibitor of IκBα, which indirectly inhibits NF-κB), and found that LPS-induced up-regulation of NLRP11 was attenuated in THP-1 cells pretreated with BAY11-7082 but not in THP-1 cells pretreated with DMSO (Fig. 1h). We further found that up-regulation of NLRP11 was completely abrogated in MyD88 knockout (KO) THP-1 cells stimulated with LPS (Fig. 1i). These data collectively suggest that NLRP11 expression is induced by stimulation of multiple TLRs via NF-κB signalling.

Because the basal level of the NLRP11 protein is relatively low in THP-1 cells and PBMCs (Fig. 1e–g), we next evaluated the turnover rate of the NLRP11 protein using the protein synthesis inhibitor cycloheximide (CHX) and found that the half-life of Flag-tagged NLRP11 was quite short (~1 h; Fig. 1j). Both ectopic and endogenous NLRP11 protein accumulated in the presence of proteasomal inhibitor MG132 (Fig. 1j, k). These data indicate that NLRP11 is a short-life protein, which is rapidly degraded via the proteasomal pathway.

To explore the cellular localization of NLRP11, we over-expressed GFP-tagged NLRP11 in HeLa cells and observed that GFP-tagged NLRP11 localized in the cytoplasm but not in the nucleus (Supplementary Fig. 1e). Because some NOD proteins, such as NLRC5, are difficult to be detected in the nucleus under normal conditions because they shuttle quickly between the cytoplasm and nucleus, leptomycin B (LMB; which inhibits CrmA-mediated nuclear export) is often used to trap certain proteins in the nucleus[34,38,39]. Here we treated HeLa cells with LMB and found that LMB treatment did not alter the cytosolic localization of NLRP11 (Supplementary Fig. 1e). Finally, we analyzed cytoplasmic and nuclear fractions of HeLa cells transfected with Flag-NLRP11 and found that NLRP11 was localized only in the cytoplasm (Supplementary Fig. 1f). As a control, NLRC5 was found to be present in both the cytoplasm and nucleus and shuttled from the cytoplasm to nucleus after treatment with LMB. Consistent with these results, endogenous NLRP11 was localized in the cytoplasm of THP-1 cells. Moreover, treatment with LMB did not affect NLRP11 localization (Fig. 1l). Taken together, these results show that NLRP11 is a cytoplasmic protein.

**NLRP11 as a potent negative regulator of NF-κB activation.** To determine whether NLRP11 participates in TLR- and/or

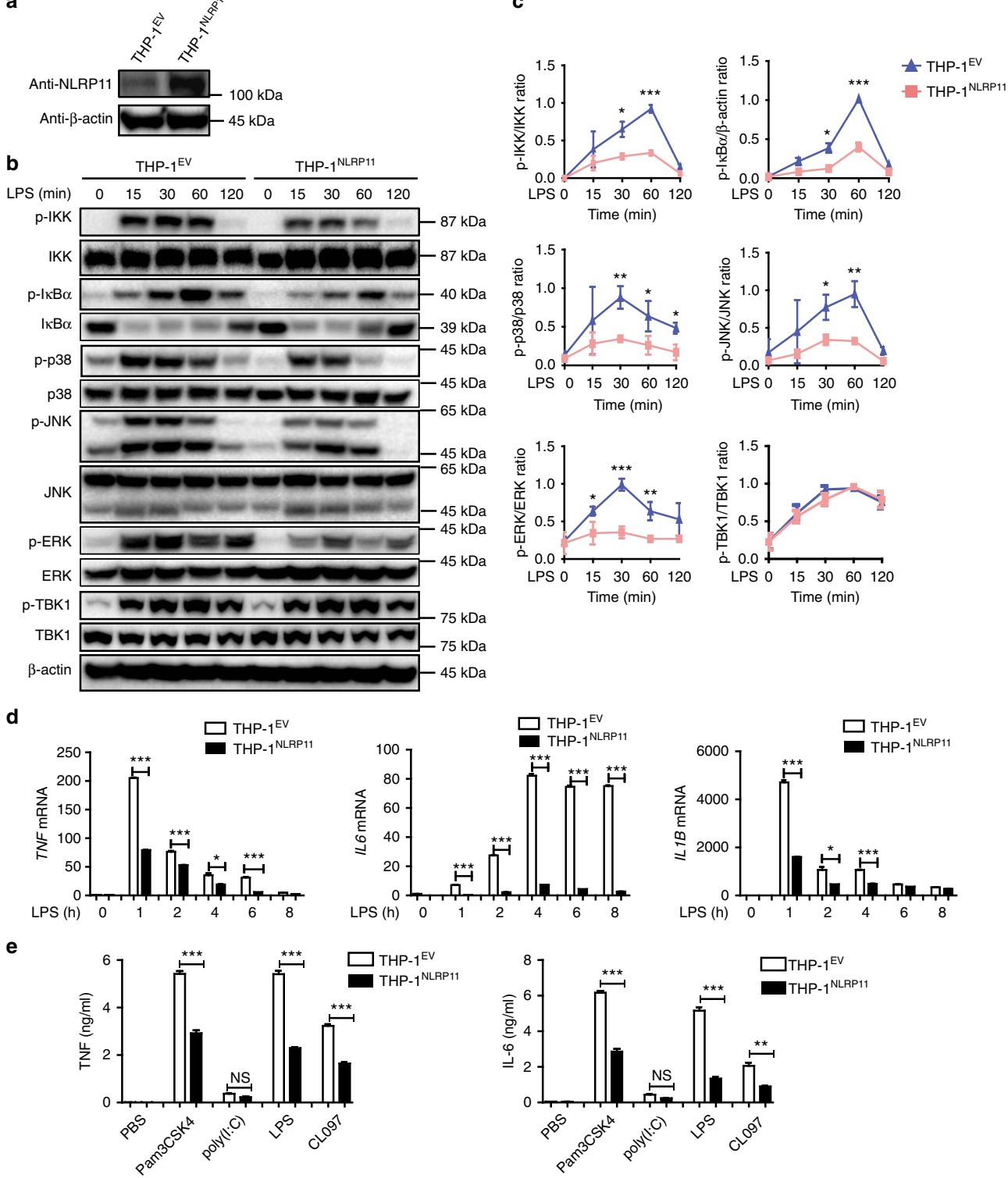

**Fig. 2** NLRP11 inhibits LPS-induced NF-κB activation. **a** Immunoblot analysis of NLRP11 in WT THP-1 cells stably expressing the empty vector (THP-1^EV) or Flag-tagged NLRP11 (THP-1^NLRP11). **b** THP-1^EV and THP-1^NLRP11 cells were stimulated with LPS (100 ng/ml) for the indicated periods, then the cell lysates were analyzed by immunoblot analysis of total and phosphorylated IKK, IκBα, TBK1, and MAPKs (p38, JNK, and ERK) (representative image). **c** Quantitative comparison of signalling activation between THP-1^EV and THP-1^NLRP11 cells by density scanning of the blots in three independent experiments. **d** THP-1^EV and THP-1^NLRP11 cells were stimulated with LPS (100 ng/ml) for the indicated periods, then analyzed by real-time PCR for *TNF*, *IL6*, and *IL1B* transcription; the results were normalized to the expression of *ACTB* (encoding β-actin) and are presented relative to those of untreated cells. **e** THP-1^EV and THP-1^NLRP11 cells were stimulated with LPS (100 ng/ml), poly(I:C) (10 μg/ml), Pam3CSK4 (100 ng/ml), or CL097 (1 μg/ml) for 24 h, then the supernatants were analyzed by enzyme-linked immunosorbent assays (ELISAs). Data **a**, **b** are representatives of three independent experiments with similar results. Data **c** are combined from three individual experiments and presented as means ± SEM. Data **d**, **e** are expressed as mean ± SEM of combined from three independent experiments with triplicate. *$P < 0.05$, **$P < 0.01$, and ***$P < 0.001$ using Student's *t*-test

cytokine-mediated NF-κB signalling, we generated an NLRP11-overexpressing THP-1 cell line (THP-1$^{NLRP11}$) (Fig. 2a, Supplementary Fig. 2a) and found that ectopic expression of NLRP11 markedly decreased LPS-induced phosphorylation of IKK, IκBα, and MAPK kinases including p38, JNK, and ERK but not TBK1

(Fig. 2b, c, Supplementary Fig. 2b). However, overexpression of NLRP11 had no effect on the activation of NF-κB and MAPKs in response to TNF treatment (Supplementary Fig. 2c). Real-time PCR analysis showed that ectopic expression of NLRP11 decreased LPS-induced expression of cytokines including TNF,

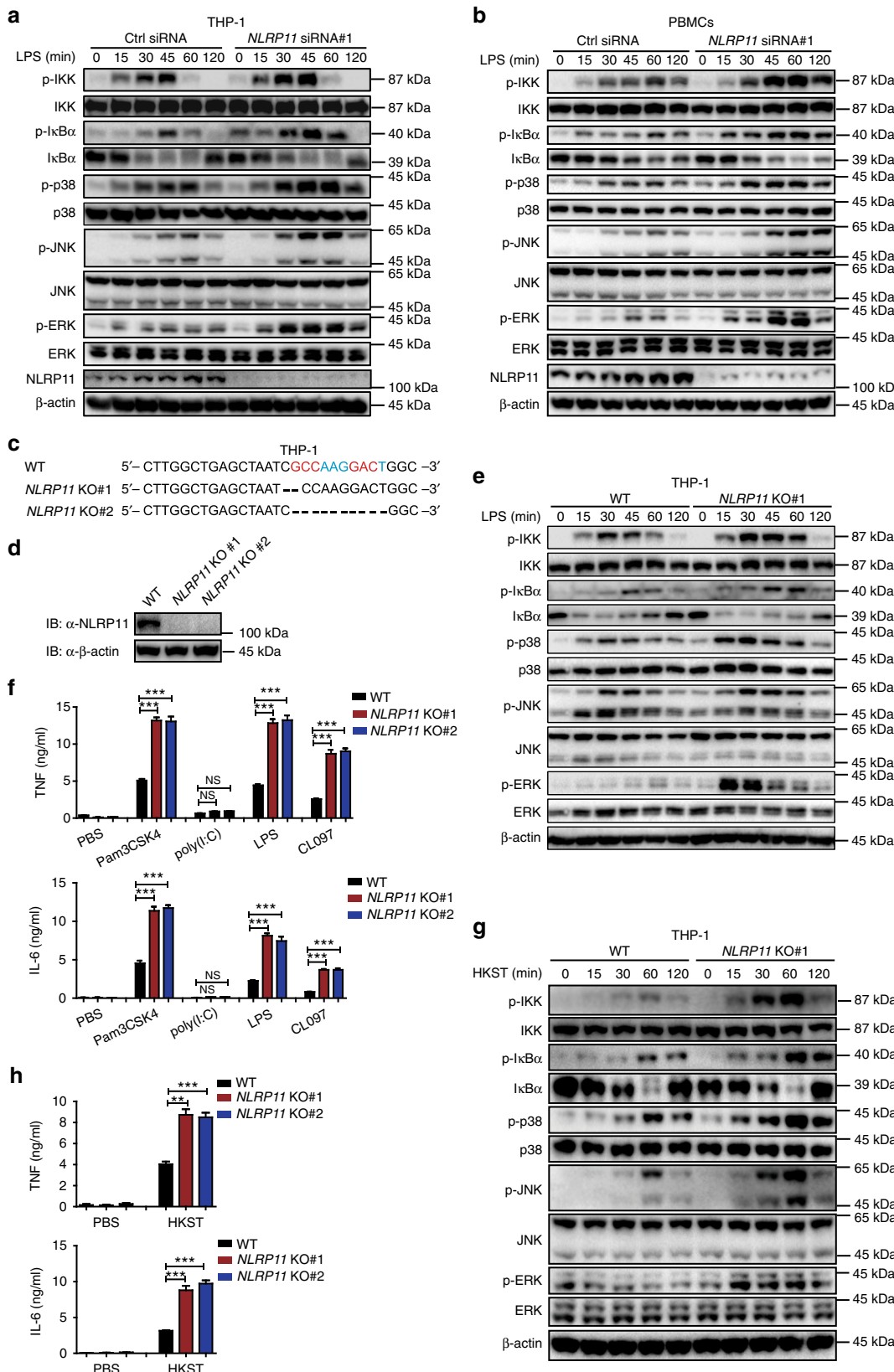

IL-6, and IL-1β (Fig. 2d). Furthermore, NLRP11 strongly inhibited the production of TNF and IL-6 in THP-1 cells stimulated with TLR agonists including LPS (a TLR4 ligand), Pam3CSK4 (a TLR2 ligand), and CL097 (a TLR7 ligand), but had little effect by poly(I:C) (a TLR3 ligand) treatment (Fig. 2e). Taken together, these results indicate that NLRP11 is a negative regulator of NF-κB activation induced by different TLR agonists.

**NLRP11 deficiency enhances TLR ligand-induced NF-κB signalling.** We next tested whether endogenous NLRP11 is required for negative regulation of LPS-triggered signalling under physiological conditions. To this end, we constructed three NLRP11-specific small interfering RNAs (siRNAs) that targeted different sites within NLRP11 mRNA. All three NLRP11-specific siRNAs efficiently downregulated NLRP11 at both mRNA and protein levels in THP-1 cells (Supplementary Fig. 3a, b). We next determined that knockdown of NLRP11 significantly enhanced LPS-induced phosphorylation of IKK, IκBα, p38, JNK, and ERK in THP-1 cells and THP-1-derived macrophages (Fig. 3a, Supplementary Fig. 3c, d). We obtained the similar results in PBMCs (Fig. 3b, Supplementary Fig. 3e). Furthermore, knockdown of NLRP11 in THP-1 cells resulted in higher expression of several proinflammatory cytokines including TNF, IL-6, and IL-1β after challenged with LPS (Supplementary Fig. 3f). Accordingly, the production of TNF and IL-6 was markedly increased by the knockdown of NLRP11 (Supplementary Fig. 3g).

To confirm these findings, we generated NLRP11 knockout (KO) THP-1 cells via the CRISPR-Cas9 approach. The deletion of NLRP11 was confirmed by genomic sequencing and immunoblotting (Fig. 3c, d). As shown in Fig. 3e, we found that activation of NF-κB and MAPKs was greatly enhanced in NLRP11 KO cells than in WT THP-1 cells. Consistently, the deficiency of NLRP11 significantly increased the production of TNF and IL-6 in response to Pam3CSK4, LPS, and CL097, but had little effect by poly(I:C) treatment (Fig. 3f). Furthermore, we examined the function of NLRP11 in response to Heat Killed Salmonella typhimurium (HKST, a TLR2, TLR4 and TLR5 agonist), and found that NLRP11 deficiency promoted the activation of NF-κB and MAPKs, and subsequently enhanced the production of TNF and IL-6 in THP-1 cells (Fig. 3g, h). Consistently, knockdown of NLRP11 in Rhesus macaca monkey PBMCs increased the expression of proinflammatory cytokines such as TNF, IL-6, and IL-1β (Supplementary Fig. 3h, i). In summary, these results suggest that NLRP11 is a conserved negative regulator of TLR signalling in primate cells.

**NLRP11 specifically interacts with TRAF6 but not TRAF2.** To further explore the regulatory mechanism of NLRP11, we transfected 293T or 293T/TLR4 cells with a NF-κB luciferase reporter vector and increasing concentrations of NLRP11, then stimulated the cells with LPS, IL-1β, or TNF. We found that NLRP11 specifically inhibited NF-κB activation induced by LPS and IL-1β but not TNF in a dose-dependent manner (Fig. 4a). Considering that TNF- and LPS- or IL-1β-triggered signalling

pathways harness different adaptors such as TRAF2 and TRAF6 to induce NF-κB activation, respectively, we examined which molecule NLRP11 can target in the regulation of NF-κB signalling. We found that NLRP11 markedly inhibited NF-κB activation in a dose-dependent manner in response to an activating signal from MyD88, IRAK1, and TRAF6 but not through TRAF2, IKKα, or p65 (Fig. 4b). We also found that NLRP11 did not inhibit TBK1-induced IFN-β activation (Fig. 4b). These results indicated that NLRP11 might specifically target TRAF6 to attenuate LPS- but not TNFα−induced NF-κB activation (Fig. 4c). We next confirmed that ectopic HA-NLRP11 interacts with Flag-TRAF6 but not Flag-TRAF2 (Fig. 4d). Furthermore, we found that endogenous NLRP11 weakly interacted with TRAF6 under normal conditions, and this interaction between NLRP11 and TRAF6 was markedly enhanced upon challenge with LPS in both THP-1 cells and THP-1-derived macrophages (Fig. 4e, f). Finally, we got a similar result in PBMCs (Fig. 4g). Collectively, our results show that NLRP11 specifically associated with TRAF6 but not TRAF2.

NLRP11 belongs to a group of NLR family members that contains a tripartite structure consisting of an N-terminal pyrin domain (PYD), a central nucleotide-binding NACHT (NOD) domain, and a C-terminal leucine-rich repeat (LRR) domain (Fig. 4h). To characterize the region in NLRP11 required for its binding to TRAF6, we generated three truncated mutants of NLRP11 and assessed their interaction with TRAF6. A coimmunoprecipitation assay showed that TRAF6 interacted with the C-terminal LRR domain but not the PYD or NOD domain of NLRP11 (Fig. 4i). Accordingly, NLRP11's LRR domain also inhibited TRAF6-induced NF-κB activity (Fig. 4j). Taken together, these results revealed that NLRP11 inhibited NF-κB signalling by targeting TRAF6 through its LRR domain.

**NLRP11 promotes ubiquitination and degradation of TRAF6.** We next sought to identify the mechanism by which NLRP11 prohibits NF-κB activation by targeting TRAF6. When we co-transfected 293T cells with NLRP11 and TRAF6, we found that NLRP11 promoted the degradation of ectopic TRAF6 in a dose-dependent manner (Fig. 5a). As a control, NLRP11 had no effect on the protein level of ectopic TRAF2 (Fig. 5a). Moreover, overexpression of NLRP11 in THP-1 cells resulted in the destabilization of endogenous TRAF6 with or without LPS stimulation (Fig. 5b). In contrast, real-time PCR data showed that TRAF6 mRNA abundance was not altered by NLRP11 overexpression in THP-1 cells both before and after LPS treatment; these data suggested that the downregulation of TRAF6 took place at the protein level, not at the mRNA level (Fig. 5c). We next found that NLRP11 deficiency markedly increased the protein level of TRAF6 in THP-1 cells (Fig. 5d). To confirm the effects of NLRP11 on TRAF6 stability in primary cells under physiological conditions, we silenced NLRP11 expression by NLRP11 siRNA in PBMCs and found that the knockdown of NLRP11 greatly enhanced the abundance of TRAF6 both before and after LPS stimulation (Fig. 5e). Collectively, these results indicate that

**Fig. 3** NLRP11 KD or KO enhances IKK and MAPK signaling. **a**, **b** Immunoblot analysis of total and phosphorylated IKK, IκBα, and MAPKs (p38, JNK, and ERK) in THP-1 cells **a** or PBMCs **b** transfected with the indicated siRNA and stimulated with LPS for indicated periods. **c**, **d** Sequence **c** and immunoblot analysis **d** of cell extracts from WT and NLRP11 knockout (KO) THP-1 cells with the indicated antibodies. **e** WT or NLRP11 KO THP-1 cells were stimulated with LPS for indicated periods, and then WCLs were subjected to immunoblot analyzes with the indicated antibodies. **f** WT or NLRP11 KO THP-1 cells were stimulated with Pam3CSK4 (100 ng/ml), poly(I:C) (10 μg/ml), LPS (100 ng/ml), or CL097 (1 μg/ml) for 24 h before the supernatants were collected. The production of TNF and IL-6 was measured by ELISA. **g** WT and NLRP11 KO THP-1 cells were infected with HKST with indicated time, and the whole-cell lysates were subjected to immunoblot analyzes with indicated antibodies. **h** WT or NLRP11 KO THP-1 cells were exposed to HKST for 24 h and the production of TNF and IL-6 was measured by ELISA. Experimental data **a**, **b**, **d**, **e**, **g** are representatives of three experiments with similar results. Data **f**, **h** are presented as mean ± SEM of combined from three individual experiments with triplicate. *$P < 0.05$, **$P < 0.01$, and ***$P < 0.001$ compared to control (Student's $t$-test)

NLRP11 negatively regulated the TLR signalling pathway by targeting TRAF6 for degradation.

We next investigated whether the degradation of TRAF6 mediated by NLRP11 was dependent on the ubiquitin-proteasome pathway or lysosomal pathway. The NLRP11-mediated destabilization of TRAF6 was reversed by the proteasomal inhibitor MG132 but not by lysosomal or autophagic inhibitors, such as $NH_4Cl$ and 3-MA, suggesting that the downregulation of the TRAF6 protein by NLRP11 was dependent on the ubiquitin-proteasome pathway (Fig. 5f). K48-linked

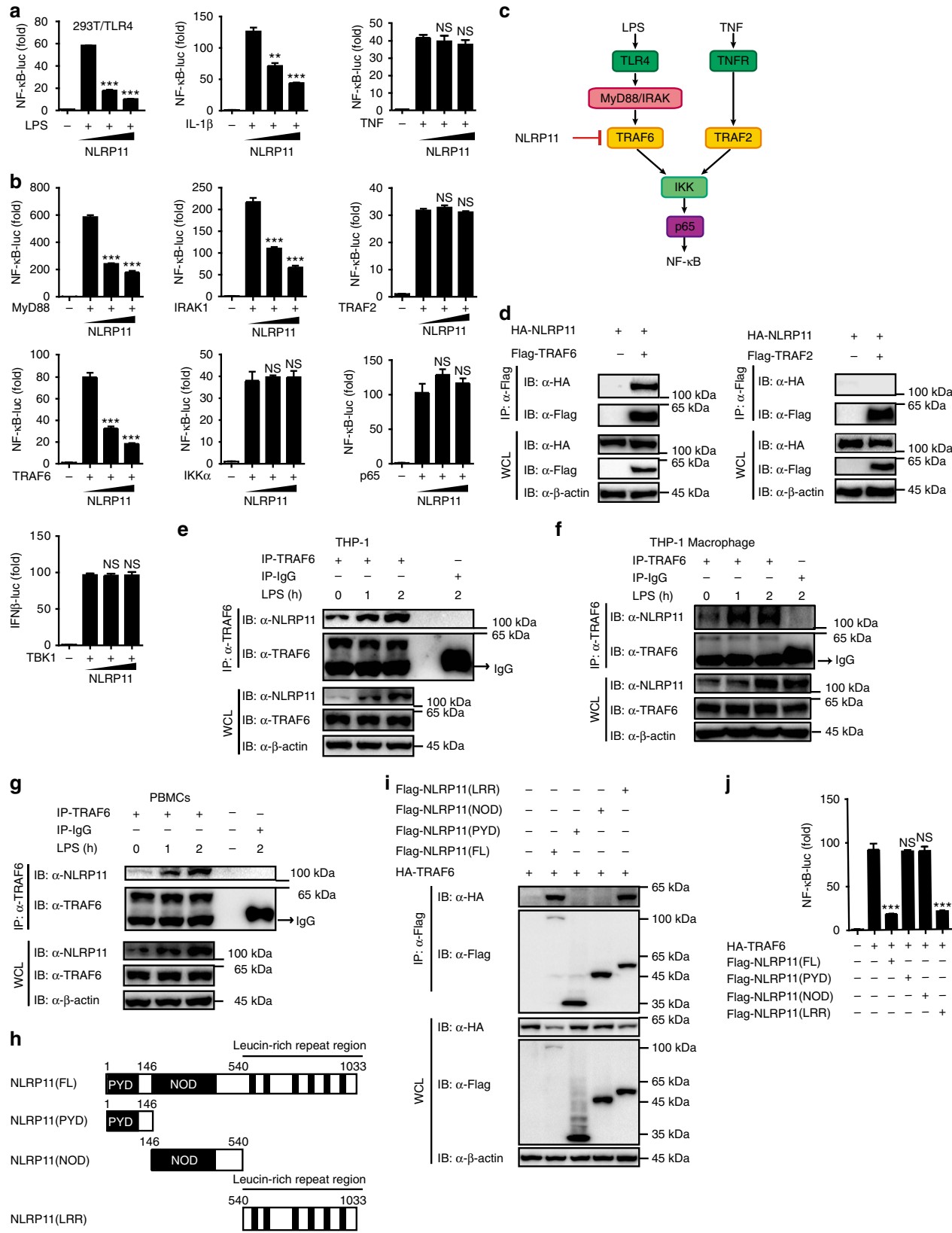

ubiquitination is typically associated with degradation via the proteasome pathway. We found that NLRP11 promoted K48-linked ubiquitination but not K63-linked ubiquitination of TRAF6 (Fig. 5g, h). Consistently, K48-linked ubiquitination of endogenous TRAF6 was detected in WT THP-1 cells in response to LPS, whereas the ubiquitination was impaired in *NLRP11* KO THP-1 cells even though endogenous TRAF6 protein abundance was increased (Fig. 5i). These results suggest that NLRP11 promoted K48-linked poly-ubiquitination of TRAF6, thereby leading to its degradation via the proteasome pathway.

**Lysines required for K48-linked ubiquitination of TRAF6**. TRAF6 consists of an N-terminal RING finger domain, a series of zinc fingers, an α-helical coiled-coil domain, and a C-terminal MATH domain (Supplementary Fig. 4a). To identify the ubiquitination sites on TRAF6, we generated TRAF6 truncation mutants and tested which domains of TRAF6 participate in the association with NLRP11. We found that NLRP11 interacted with the MATH and ZF-MATH domains of TRAF6 (Supplementary Fig. 4b). We next sought to determine which domain of TRAF6 undergoes K48-linked poly-ubiquitination. Like WT TRAF6 protein, MATH and ZF-MATH domains of TRAF6 but not other TRAF6 mutants lacking the MATH domain were ubiquitinated with K48 linkage (Fig. 6a). According to above results, we reasoned that there might be one or more lysine residues in the MATH domain responsible for the K48-linked ubiquitination of TRAF6. To identify the lysine (K) residues of TRAF6 to which ubiquitin is attached, we carried out a systematic lysine (K) to arginine (R) mutation screening (Fig. 6b). When the eight lysines (K489, K469, K453, K388, K384, K371, K365, and K356) in TRAF6 were all substituted with arginines (8KR), the K48-linked ubiquitination of the 8KR mutant was abrogated even in the presence of NLRP11 (Fig. 6c). We then re-substituted the arginine residues in the 8KR mutant with lysines. It was observed that the ubiquitination of TRAF6 reappeared in the R356K, R365K, and R371K mutants (Fig. 6c). To exclude the possibility of any artifacts, we generated single-site mutations (K356R, K365R, and K371R) and a triple K-to-R mutation (3KR). As shown in Fig. 6d, single-site mutations had little effect on the K48-linked poly-ubiquitination of TRAF6, whereas the 3KR mutation markedly reduced K48-linked poly-ubiquitination of TRAF6 compared with WT TRAF6. Consistently, NLRP11 promoted the endogenous K48-linked poly-ubiquitination of WT TRAF6, but had little effect on 3KR mutant in TRAF6 KO 293T cells. Moreover, NLRP11 did not affect K63-linked poly-ubiquitination of WT and 3KR mutants (Fig. 6e, Supplementary Fig. 4c). Thus, these data indicate that these three lysines are the major K48-linked ubiquitination sites on TRAF6.

We next determined whether these three ubiquitination sites (K356, K365, and K371) in TRAF6 are indeed functionally involved in the NLRP11-mediated degradation of TRAF6. We assessed the degradation of TRAF6 in *TRAF6* KO 293T cells reconstituted with either TRAF6 WT or TRAF6 mutants (K356R, K365R, K371R, and 3KR; Fig. 6f). Overexpression of NLRP11 promoted the degradation of WT TRAF6 or single-point mutants of TRAF6 including K356R, K365R, and K371R. In contrast, the 3KR TRAF6 mutant was completely resistant to NLRP11-mediated degradation, suggesting that all three sites may cooperatively regulated by NLRP11 (Fig. 6f). To verify the functional importance of the ubiquitination of TRAF6 at K356, K365, and K371, we examined the activation of NF-κB in a luciferase reporter assay of *TRAF6* KO 293T cells. As expected, we found that NLRP11 failed to inhibit NF-κB activation induced by the 3KR TRAF6 mutant, whereas NF-κB activation induced by WT TRAF6 or by other TRAF6 mutants was strongly inhibited by NLRP11 (Fig. 6g). Collectively, these results suggest that the three lysines (K356, K365, and K371) are critical residues for NLRP11-mediated K48-linked ubiquitination and degradation of TRAF6.

**NLRP11 recruits RNF19A to degrade TRAF6**. The above results showed that NLRP11 promoted the K48-linked ubiquitination and degradation of TRAF6. Nevertheless, NLRP11 is not an E3 ubiquitin ligase. It has been reported that TRIM38 catalyzes K48-linked ubiquitination of TRAF6 and subsequently leads to its degradation in the mouse RAW264.7 cell line[22]. Therefore, we hypothesized that NLRP11 may function as an adaptor to recruit TRIM38 to degrade TRAF6 in human cells. To test this hypothesis, we generated three siRNA targeted human TRIM38 and the knockdown efficiency was detected by real-time PCR (Supplementary Fig. 5a). Knockdown of TRIM38 had no effect on the expression of TRAF6 in THP-1 derived-macrophages (Supplementary Fig. 5b). In addition, TRIM38 did not promote NLRP11-mediated degradation of TRAF6 (Supplementary Fig. 5c). Thus, we proposed that the regulatory mechanism of TRAF6 stability might be different between human and mice. We next screened a sub-library containing ~900 lentivirus-based short hairpin RNAs (shRNAs) by co-transfecting NLRP11 and TRAF6 with each shRNA clone into 293T cells (Supplementary Table 3), and found that a knockdown of RNF19A almost completely reversed the NLRP11-mediated inhibition of TRAF6-induced NF-κB activation (Supplementary Fig. 5d). To substantiate this finding, we further selected three shRNAs against RNF19A and three other E3 ligase shRNAs as controls and the knockdown efficiency were confirmed by real-time PCR (Supplementary Fig. 5e). Specifically, we found that knockdown of endogenous RNF19A by shRNA markedly abrogated the ability

**Fig. 4** NLRP11 specifically associates with TRAF6 but not TRAF2. **a** 293T-TLR4 cells or 293T cells were co-transfected with a NF-κB and TK-Renilla reporter along with increasing amounts of NLRP11 for 24 h and then left untreated or stimulated with LPS or TNF for 6 h before a luciferase assay. NF-κB promoter-driven luciferase activity was measured and normalized to the *Renilla* luciferase activity. **b** Luciferase activity in 293T cells transfected with the NF-κB or IFN-β luciferase reporter, together with a vector encoding MyD88, IRAK1, TRAF2, TRAF6, IKKα, p65, or TBK1, along with the empty vector or with increasing amounts of a vector encoding NLRP11. The results are presented relative to *Renilla* luciferase activity. **c** Schematic overview of TLR4- and TNFR-mediated NF-κB activation, regulated by NLRP11. **d** Immunoprecipitation and an immunoassay of lysates of 293T cells transfected with a vector expressing HA-tagged NLRP11 along with the empty vector or vector encoding Flag-tagged TRAF2 or TRAF6. **e–g** THP-1 cells **e**, THP-1-derived macrophages **f**, or PBMCs **g** were stimulated with LPS for the indicated periods. The cell lysates were subjected to immunoprecipitation with an anti-TRAF6 antibody or control IgG, followed by immunoblotting with an anti-NLRP11 or anti-TRAF6 antibody. **h** A structural diagram of NLRP11 as well as schematic representation of Flag-tagged truncation mutants of NLRP11. The numbers indicate amino acid positions. **i** 293T cells were transfected with HA-tagged TRAF6 and Flag-tagged NLRP11 (FL) or NLRP11 truncation mutants. The cell lysates were subjected to immunoprecipitation with anti-Flag antibodies and immunoblotted with the indicated antibodies. **j** Luciferase activity in 293T cells transfected with an NF-κB luciferase reporter, together with a vector encoding TRAF6, along with the empty vector or with vectors encoding NLRP11 or its mutants. The results are presented relative to *Renilla* luciferase activity. Data **a**, **b**, and **j** are expressed as mean ± SEM of combined from three independent experiments with triplicate; *P < 0.05, **P < 0.01, and ***P < 0.001 compared to the same treatment in control cells (Student's *t*-test). Data **d–g** and **i** are representatives of three independent experiments with similar results

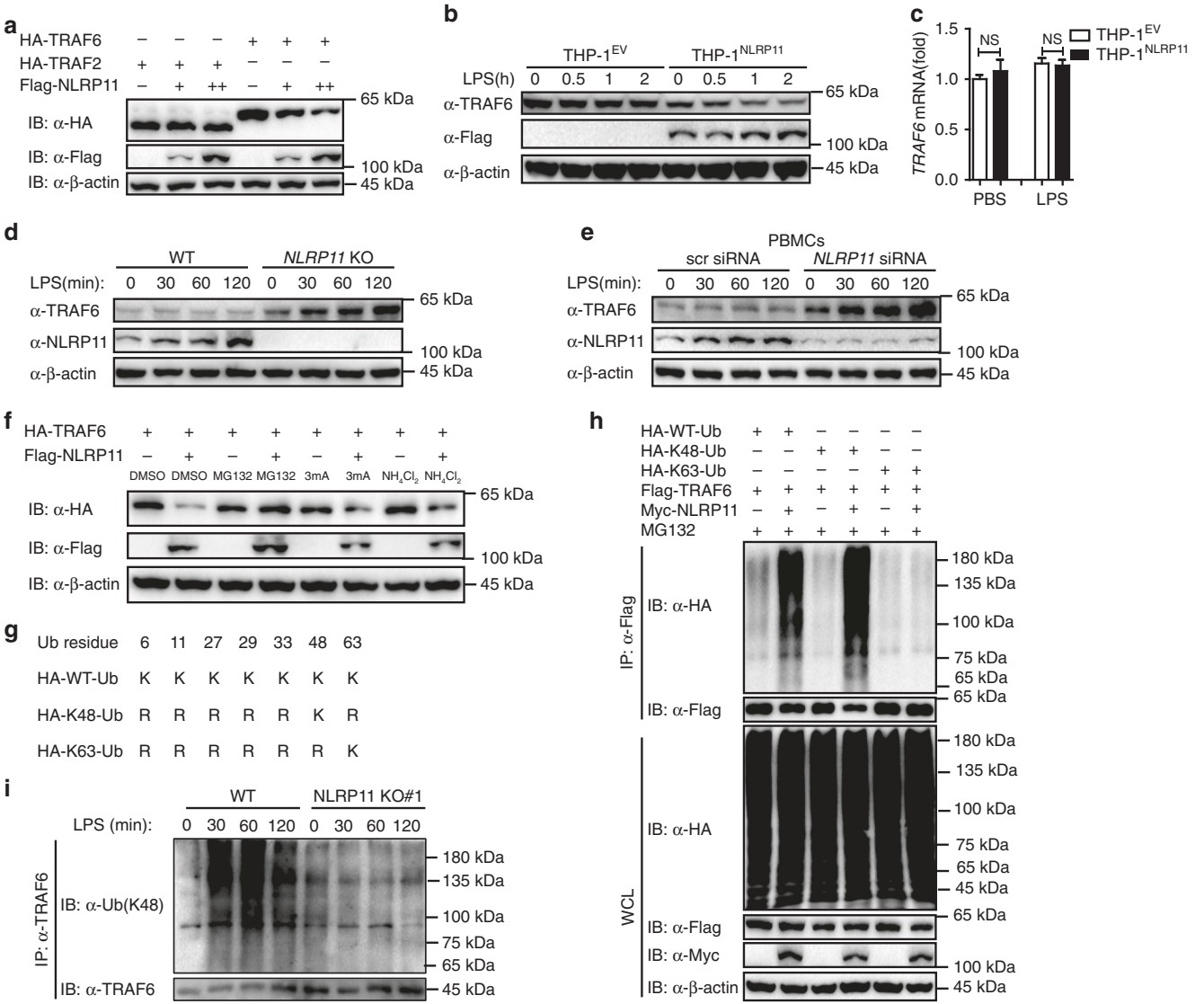

**Fig. 5** NLRP11 promotes ubiquitination and degradation of TRAF6. **a** Immunoblot analysis of 293T cells transfected with HA-tagged TRAF6 or TRAF2 and the empty vector or increasing amounts of the vector encoding Flag-tagged NLRP11. **b** Immunoblot analysis of TRAF6 in WT THP-1 cells stably expressing the empty vector (THP-1^EV) or Flag-tagged NLRP11 (THP-1^NLRP11), treated with LPS at different time points. **c** THP-1 cells as in **b** were incubated with or without LPS, and then total RNA was isolated to measure the transcription levels of *TRAF6* by real-time PCR. **d** An immunoassay of endogenous TRAF6 in wild type (WT) or *NLRP11* knockout (KO) THP-1 cells stimulated with LPS for the indicated periods. **e** PBMCs were transfected with *NLRP11*-specific siRNA or control siRNA for 48 h, and the cells were treated with LPS for the indicated periods and subjected to immunoblotting with the indicated antibodies. **f** Immunoblot analysis of extracts from 293T cells transfected with HA-TRAF6 and FLAG-NLRP11 or the control vector for 24 h, then treated for 6 h with 10 μM MG132, 10 mM 3-MA, 20 mM NH₄Cl, or DMSO. **g** A schematic diagram of ubiquitin and its mutants. **h** 293T cells were transfected with Flag-TRAF6 together with the empty vector or a vector encoding Myc-NLRP11 as well as with a vector encoding HA-WT-ubiquitin (Ub) or its mutants (HA-K48-Ub or HA-K63-Ub), and then treated with MG132 (10 μM) for 6 h. WCLs were subjected to denaturing immunoprecipitation with anti-Flag beads and analyzed by immunoblotting with the indicated antibodies. **i** WT and *NLRP11* KO THP-1 cells were stimulated with LPS for the indicated periods, and the K48-linked ubiquitination of endogenous TRAF6 was determined by immunoprecipitation with an anti-TRAF6 antibody and immunoblotting with the indicated antibodies. Data **a**, **b** and **d**–**i** are representative three individual experiments with similar results. Data **c** are presented as mean ± SEM of combined from three independent experiments with triplicate. NS, not significant. Significance was assessed using Student's *t*-test

of NLRP11 to inhibit TRAF6-mediated NF-κB activation and to induce the degradation of TRAF6; the knockdown of other E3 ubiquitin ligases such as DTX4 or RNF7 did not have these effects (Fig. 7a). Moreover, knockdown of RNF19A in PBMCs resulted in higher expression of proinflammatory cytokines including TNF, IL-6, and IL-1β in response to LPS treatment (Fig. 7b, Supplementary Fig. 5f), which indicated that RNF19A plays a key role in the NF-κB signalling pathway. Consistently, NLRP11 failed to inhibit the LPS-induced TNF and IL-6 production in THP-1 cells with silencing of RNF19A (Fig. 7c). Thus, these

results indicate that the E3 ligase RNF19A is necessary in the inhibition of NF-κB signalling mediated by NLRP11.

To confirm the involvement of RNF19A in NLRP11-mediated degradation of TRAF6, we generated *RNF19A* KO 293 T cells (Fig. 7d, e). As expected, NLRP11 failed to induce the degradation of TRAF6 and to inhibit TRAF6-induced NF-κB activation in *RNF19A* KO cells (Fig. 7f, Supplementary Fig. 5g). Moreover, RNF19A deficiency completely impaired the ability of NLRP11 to induce K48-linked ubiquitination of TRAF6 (Fig. 7g). Altogether, these results suggest that NLRP11-mediated K48-linked

ubiquitination and degradation of TRAF6 are dependent on E3 ubiquitin ligase RNF19A.

We next found that RNF19A alone could not induce TRAF6 degradation and inhibition of NF-κB activity induced by TRAF6 in 293T cells. Rather, co-expression of RNF19A and NLRP11

resulted in more TRAF6 degradation than did expression of NLRP11 alone (Fig. 7h). Accordingly, RNF19A overexpression markedly augmented the inhibition of TRAF6-induced NF-κB activation by NLRP11 (Supplementary Fig. 5h). We thus reasoned that NLRP11 functions as an adaptor to recruit the E3 ubiquitin

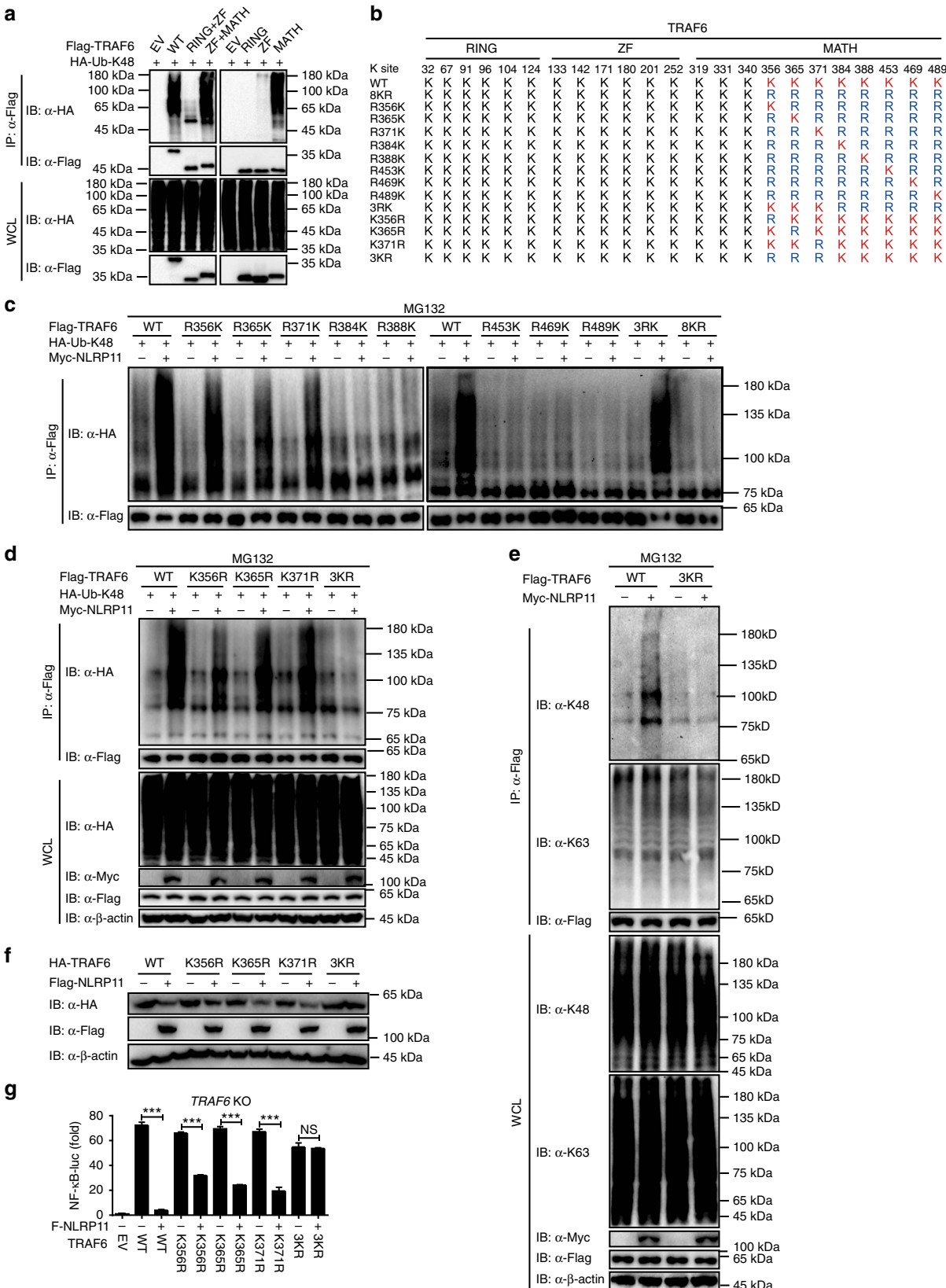

ligase RNF19A to degrade TRAF6. To test this hypothesis, we conducted coimmunoprecipitation experiments to analyze the interaction between RNF19A and TRAF6 in the presence or absence of NLRP11 and found that TRAF6 associated with RNF19A only in the presence of NLRP11 (Fig. 7i). Additionally, we confirmed the interaction between NLRP11 and RNF19A (Fig. 7j). Furthermore, NLRP11 deficiency abrogated this interaction between RNF19A and TRAF6 in response to LPS in THP-1 cells (Fig. 7k). Collectively, these results suggest that NLRP11 functions as an adaptor to recruit E3 ubiquitin ligase RNF19A to TRAF6 and facilitates the K48-linked ubiquitination of TRAF6 for its subsequent degradation.

## Discussion

The TRAF protein family plays a pivotal role in the TLR-mediated inflammatory response, thus their activities must be strictly controlled to maintain immunological homeostasis. There are seven members of TRAF proteins (TRAF1–7) that are characterized by the presence of the TRAF domain, which is responsible for the interactions between TRAF proteins and the upstream receptors or downstream signalling molecules. All TRAFs, except TRAF1, contain an N-terminal RING finger domain that may serve as an E3 ubiquitin ligase to catalyze ubiquitin ligation to target proteins. It has been reported that NLRs and TRAF proteins are functionally related and may form a complex called "TRAFasome," to regulate immune responses[29]. For example, NLRX1 interacts with TRAF6 to negatively regulate NF-κB signalling via a yet-to-be defined mechanism, and NLRC3 physically binds to TRAF6 and removes K63-linked poly-ubiquitination of TRAF6 in response to LPS treatment[29–31]. NLRP12 also inhibits noncanonical NF-κB activation by targeting NIK for degradation possibly through association with TRAF3[40]. Thus, it is of great interest to identify whether other NLRs participate in TRAFasome formation to regulate immune responses. Here our study provided several lines of evidence that NLRP11 is a novel inhibitor of NF-κB signalling by controlling TRAF6 activity in human cells. First, like many other negative regulators, NLRP11 was induced by TLR ligands, such as Pam3CSK4 and LPS. Second, NLRP11 associated with TRAF6, and this interaction was enhanced after LPS treatment. Third, whereas exogenous NLRP11 inhibited the activation of NF-κB and MAPK signalling and subsequent production of proinflammatory cytokines, NLRP11 deficiency promoted inflammatory responses. We further found that NLRP11 specifically inhibits TRAF6-dependent NF-κB activation. Thus, our findings identified a previously unrecognized role for NLRP11 in the attenuation of TLR signalling by targeting TRAF6 for degradation.

Ubiquitination is a critical modification involved in the regulation of innate and adaptive immune responses[14,41–43]. Modification of TRAF6 by different types of ubiquitin chains has been identified as a key step to orchestrate TRAF6-mediated NF-κB and MAPK activation. It has been reported that K63-linked ubiquitination of TRAF6 is necessary for the activation of TLR signalling[10,13,41]. On the other hand, K48-linked ubiquitination of

TRAF6 promotes proteasomal degradation of TRAF6, thereby limiting an innate immune response. For instance, TRIM38 has been reported to negatively regulate TLR-triggered NF-κB activation by mediating degradation of TRAF6 via the ubiquitin-proteasome pathway in RAW264.7 cells. In contrast, another study showed that the protein level of TRAF6 is comparable in TRIM38[+/+] and TRIM38[−/−] bone marrow-derived dendritic cells before and after LPS treatment[22,44]. Here, our results show that that TRIM38 did not promote NLRP11-mediated degradation of TRAF6 in human cells. Thus, we speculate that human and mice may harness different regulatory mechanisms of TRAF6 degradation, and these work emphasized the necessary for multiple and precise mechanisms to control TRAF6 signalling.

In this study, we demonstrated that NLRP11 promotes K48-linked ubiquitination at three lysine residues of TRAF6 for its proteasomal degradation and emphasized the necessary for multiple and precise modifications of TRAF6 signalling. Our previous report indicated that another NLR protein, NLRP4, serves as an adaptor to recruit the E3 ubiquitin ligase DTX4 to catalyze K48-linked poly-ubiquitination of TBK1, which leads to the degradation of TBK1 and inhibits type I interferon signalling[45]. Similar to NLRP4, NLRP11 itself is not an E3 ubiquitin ligase. There must be additional E3 ligases involved in this process. Among the RING finger domain-containing E3 ligases, we identified RNF19A as the enzyme that affects TRAF6 turnover. RNF19A deficiency abrogated the ability of NLRP11 to induce the degradation of TRAF6. In addition, NLRP11 could not promote K48-linked ubiquitination in RNF19A KO cells. Elimination of NLRP11 also abrogated the ability of RNF19A to interact with TRAF6. Thus, our data provide insights into the molecular mechanisms by which NLRP11 recruits E3 ligase RNF19A to degrade TRAF6 in the TLR-mediated signalling pathway.

Several NLRPs (like NLRP2, 4, 5, 7, 9, 11, and 14) have been shown to specifically or preferentially expressed in mammalian oocyte[35]. NLRP14 was recently identified as a germ-cell-specific inhibitor of cytosolic nucleic acid sensing to promote fertilization[46], suggesting that controlling innate immune response is crucial to maintain proper immunological homeostasis in germline. In addition, the activation of NLRP3 inflammasome and mitosis are mutually exclusive events mediated by NEK7[47,48]. Here we demonstrated that NLRP11 was highly expressed in testis and ovary and paly pivotal roles in attenuating TLR signalling, implying that NLRP11 may also have special functions between innate immune response and fertilization development.

On the basis of our findings discussed above, we propose the following working model to explain how NLRP11 negatively regulates TLR-mediated NF-κB and MAPK signalling pathways (Fig. 8). NLRP11 expression is upregulated in a NF-κB dependent manner, thus constituting a negative feedback loop to control NF-κB activity. Once upregulated, NLRP11 functions as an adaptor to recruit ubiquitin ligase RNF19A to promote K48-linked ubiquitination and proteasomal degradation of TRAF6, thereby resulting in decreased production of proinflammatory cytokines. Overall, our findings have uncovered a previously unrecognized

**Fig. 6** NLRP11 promotes K48-linked poly-ubiquitination of TRAF6. **a** 293T cells transfected with the indicated plasmids were subjected to immunoprecipitation and then immunoblotted with the indicated antibodies. **b** A schematic diagram of TRAF6 and its mutants. **c, d** 293T cells were transfected with Myc-NLRP11, HA-K48-Ub, and Flag-TRAF6 RK **c** or KR **d** mutants with the indicated combinations for 24 h and then treated with MG132 (10 μM) for 6 h. WCLs were subjected to denaturing immunoprecipitation with anti-Flag beads and then analyzed by immunoblotting with an anti-HA or anti-Flag antibody. **e** TRAF6 KO 293T cells transfected with TRAF6 WT or 3KR mutant and Myc-NLRP11 were subjected to denaturing immunoprecipitation with an anti-Flag antibody followed by immunoblotting with the indicated antibodies. All the cells were treated with 10 μM MG132 for 6 h before collecting. **f** Immunoblot analysis of 293T cells transfected with the indicated combinations of expression plasmids **g** TRAF6 KO 293T cells were transfected with the indicated plasmids together with the NF-κB luciferase reporter and pTK-Renilla reporter plasmids for 24 h before luciferase assays were performed. Data **a** and **c–f** are representatives of three independent experiments with similar results. Data **g** are shown as mean ± SEM of combined from three independent experiments with triplicate. ***P < 0.001 compared to the control group (Student's t-test)

mechanism by which NLRP11 attenuates TLR-mediated inflammatory responses and highlights NLRP11 as a potential target for therapy against inflammatory diseases and fertilization.

## Methods

**Antibodies and reagents**. Antibodies against Flag (A8592, 1:1000) and β-actin (A1978, 1:1000) were purchased from Sigma. Antibodies against HA (12013819001, 1:1000) and Myc (11814150001, 1:1000) were purchased from Roche Applied Science. Antibodies against IKK (#05-535, 1:1000) were purchased from Millipore. An antibody against NLRP11 (ab103333, 1:1000) were purchased

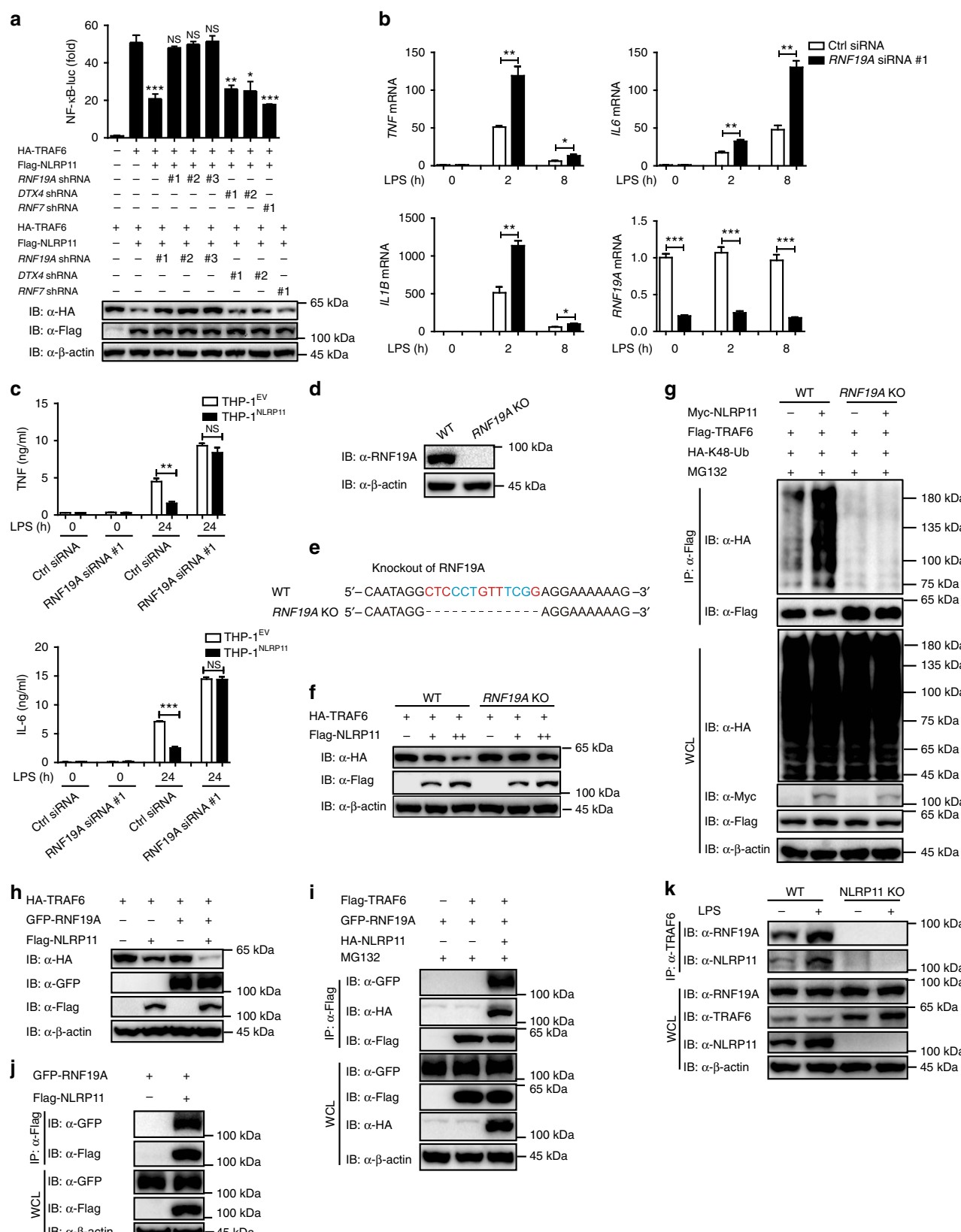

from Abcam. Antibodies against TRAF6 (sc-8409, 1:500) and RNF19A (sc-55808, 1:500) were purchased from Santa Cruz Biotechnology. Antibodies against the following proteins were purchased from Cell Signalling Tchnology: p-IKKα/β (#2697, 1:1000), p-IκBα (#9246, 1:1000), IκBα (#4814, 1:1000), p-p38 (#9211, 1:1000), p38 (#9212, 1:1000), p-JNK (#9251, 1:1000), JNK (#9252, 1:1000), p-ERK (#9101, 1:1000), ERK (#9102, 1:1000). Recombinant human TNF-α (10 ng/ml) and IL-1β (10 ng/ml) were purchased from Pepro Tech. Lipopolysaccharides (LPS) (L4391-1MG, 100 ng/ml) was purchased from Sigma. Pam3CSK4 (100 ng/ml), poly (I:C) (10 μg/ml) and HKST ($10^6$ cells per ml) were purchased from InvivoGen.

**Cell culture and transfection**. HEK293T cells (ATCC, Cat# CRL-3216) were maintained in Dulbecco's modified Eagle's medium (DMEM, Corning), whereas THP-1 cells (ATCC, Cat# TIB-202) were maintained in RPMI 1640 (Corning) in a 5% (vol/vol) $CO_2$ incubator at 37 °C. All media were supplemented with 10% (vol/vol) of fetal bovine serum (FBS), 100 U ml$^{-1}$ penicillin, 100 U ml$^{-1}$ streptomycin, and 4 nM L-glutamine. PBMCs were isolated from Buffy coats of blood from healthy donors (from Guanzhou Blood Center) by density gradient centrifugation and washed with phosphate-buffered saline (PBS, Corning), and then grown in RPMI 1640. Transfection of HEK293T cells involved Lipofectamine 2000 (Invitrogen).

**Differentiation of THP-1-derieved macrophages**. THP-1 cells were seeded in 6-well plate at the density of $1 \times 10^6$ cells per ml. THP-1 cells were differentiated to macrophages by stimulation with 60 nM PMA for 16 h, and then cultured for an additional 48 h in fresh medium prior to further treatment.

**Immunoblotting and immunoprecipitation**. HEK293T cells were transfected with the indicated combinations of plasmids and collected 24 h after transfection. For western blotting, proteins in cell lysates were separated by SDS-polyacrylamide gel electrophoresis (PAGE) in an 8–12% (wt/vol) gel and transferred to polyvinylidene difluoride membranes (Bio-Rad). Each membrane was blocked with 5% (wt/vol) nonfat skim milk in TBS-T buffer (50 mM Tris-HCl, 150 mM NaCl, 0.05% Tween 20). After three washes in TBS-T, the membrane was incubated with an appropriate antibody diluted in 5% (wt/vol) nonfat skim milk-TBS-T or 5% (wt/vol) BSA-TBS-T. For analysis of the endogenous protein, the membrane was subsequently visualized after incubating with a secondary antibody: a horseradish peroxidase-conjugated anti-mouse or anti-rabbit IgG antibody by means of ChemiDoc XRS + (Bio-Rad). For immunoprecipitation, cell lysates were prepared in low-salt buffer containing phosphatase and protease inhibitors and were incubated with the indicated primary antibody along with Protein A/G beads at 4 °C overnight with gentle shaking. Beads with anti-Flag and anti-HA antibodies were used for immunoprecipitation of proteins containing a Flag or HA tag. After five washes with low-salt buffer, the precipitated proteins were boiled with 3× SDS loading buffer (FDBio Technology) and analyzed by immunoblotting. Full length uncropped western blots are presented in Supplementary Figs. 6–17.

**Real-time PCR**. Total RNA was isolated with the TRIzol reagent (Invitrogen), and cDNA was synthesized using the EasyScript First-Strand cDNA Synthesis Super-Mix (TAKARA), in which a 500 ng RNA sample served as a template. Quantitative real-time PCR was conducted on a QuantStudio(TM) 6 Flex System (Applied Biosystems [part of Life Technologies]) with the SYBR Green qPCR SuperMix (Genestar). The results were analyzed by the comparative threshold cycle ($C_t$) quantification method, in which β-actin served as an internal control. The primers for real-time PCR are shown in Supplementary Table 1.

**RNA interference**. The cells were plated at the density of $5 \times 10^5$ per milliliter and transfected with siRNA by means of Lipofectamine-RNAi MAX (Invitrogen), and after 24–48 h, the cells were used in further experiments. The final concentration of siRNA was 10 nM, and the target siRNA sequences are described in Supplementary Table 2.

**Generation of NLRP11 KO cells by CRISPR-Cas9**. Guide RNA (gRNA) was designed using an online gRNA design tool http://crispr.mit.edu/ (by Zhang Feng lab) and was subcloned into the pLentiCRISPR V2 vector for expressing gRNA and Cas9; this vector was transfected into HEK293T cells together with two packing plasmids: vsvg and Δ8.9. The culture supernatants containing the lentivirus were collected 48 h after transfection and concentrated by ultracentrifugation before their use for infection of THP-1 cells. Infection-positive cells were selected and enriched by selection on puromycin and were assessed by a T7 endonuclease I assay. Monoclonal cells were screened by a limiting dilution assay and then confirmed by sequencing of PCR fragments and western blot analysis of cell lysates with a corresponding antibody. gRNAs used for generating the NLRP11 KO cells are listed below:

NLRP11-sgRNA:
sense: 5′-GCTTGGCTGAGCTAATCGCCA-3′;
antisense: 5′-TGGCGATTAGCTCAGCCAAGC-3′.

**An in vivo ubiquitination assay**. At 24 h post-transfection, cells were lysed with SDS lysis buffer (50 mM Tris-HCl, pH 6.8, 150 mM NaCl, 10% glycerol, 1% SDS) containing protease inhibitors. Before 10-fold dilution with dilution buffer (10 mM Tris-HCl, pH 8.0, 150 mM NaCl, 2 mM EDTA, 1% Triton X-100), the samples were boiled with SDS buffer for 4 min. After incubation at 4 °C for 30 min, the diluted lysates were centrifuged at 20,000 r.p.m. for 30 min at 4 °C, and the supernatants were subjected to immunoprecipitation with the indicated antibodies.

**The luciferase assay**. HEK293T cells were seeded in a 96-well plate ($4 \times 10^4$ per well) and transfected with the indicated plasmid and a luciferase reporter plasmid NF-luc together with pRL-TK (Renilla luciferase plasmid) as a control reporter vector. Samples were prepared in triplicate, and the empty pcDNA3.1 vector was used to equalize total DNA amounts among wells. At 24 h post-transfection, the cells were disrupted with lysis buffer (Promega), and luciferase activity was measured by the Dual-Luciferase Reporter Assay (Promega) on a Synergy 2 microplate reader (BioTek). The results were calculated by normalization of firefly luciferase activity to Renilla luciferase activity.

**Enzyme-linked immunosorbent assays**. Supernatants from cultured cells with different pretreatments were collected at indicated time points after stimulation, and concentrations of human IL-6, IL-1β, and TNF were assessed using BD OptEIA ELISA kits (BD Biosciences).

**Site-directed mutagenesis**. The template plasmid was amplified with a pair of primers containing a point mutation by means of the Q5 High-Fidelity PCR Kit (New England Biolabs), and the products were subsequently digested with 10 U of DpnI (New England Biolabs) at 37 °C for 1 h and were transfected into DH5α competent cells. Plasmids were extracted with the E.Z.N.A. Plasmid Mini Kit I (Omega).

**Statistics**. All experiments were repeated at least three times, and data are presented as mean ± SD from three independent experiments. Significance was determined by two-tailed Student's t-test in the GraphPad Prism5 software, and differences with a P-value <0.05 were considered statistically significant.

**Data availability**. The data that support the findings of this study are available from the corresponding author upon request.

---

**Fig. 7** NLRP11 recruits RNF19A to degrade TRAF6. **a** 293T cells were transfected with the indicated shRNA, an NF-κB reporter together with HA-TRAF6, Flag-NLRP11 or the control vector for 24 h, and then subjected to luciferase assay and immunoblotting analysis. **b** PBMCs were transfected with RNF19A or control siRNA for 48 h, stimulated with LPS (100 ng/ml), then analyzed by real-time PCR for TNFA, IL6, IL1B, and RNF19A expression. **c** ELISA analysis of TNF and IL-6 production in THP-1$^{EV}$ and THP-1$^{NLRP11}$ cells that were transfected with RNF19A siRNA and stimulated with LPS for 24 h. **d** Immunoblotting analysis of cell extracts from WT and RNF19A knockout (KO) 293T cells with the indicated antibodies. **e** Sequence analysis of WT and RNF19A KO 293T cells. **f** Immunoblot analysis of lysates of WT and RNF19A KO 293T cells transfected with HA-TRAF6 and the empty vector or increasing concentrations of Flag-NLRP11. **g** WT and RNF19A KO 293T cells transfected with Flag-TRAF6 and HA-K48-Ub along with the empty vector or Myc-NLRP11 were subjected to immunoprecipitation with an anti-Flag antibody and immunoblotting with the indicated antibodies. All the cells were treated with 10 μM MG132 for 6 h before collecting. **h** Immunoblot analysis of 293T cells transfected with the indicated expression plasmids. **i** 293T cells transfected with the indicated expression vectors were used for coimmunoprecipitation with anti-Flag beads followed by immunoblotting with the indicated antibodies. **j** Flag-NLRP11 together with empty vector or GFP-RNF19A were co-transfected into 293T cells. Whole-cell extracts were subjected to immunoprecipitation with an anti-Flag antibody and immunoblotted with the indicated antibodies. **k** An immunoassay of lysates of WT and NLRP11 KO THP-1 cells left untreated or treated for 60 min with LPS, analyzed by immunoprecipitation with an anti-TRAF6 antibody and immunoblot with antibodies to NLRP11 or RNF19A. Data a up panel, **b**, **c** are plotted as the mean ± SEM of three independent experiments with triplicate. *P < 0.05, **P < 0.01, and ***P < 0.001 compared to control group using Student's t-test. Data a down panel, **d**, **f**, **g–k** are representatives of three independent experiments with similar results

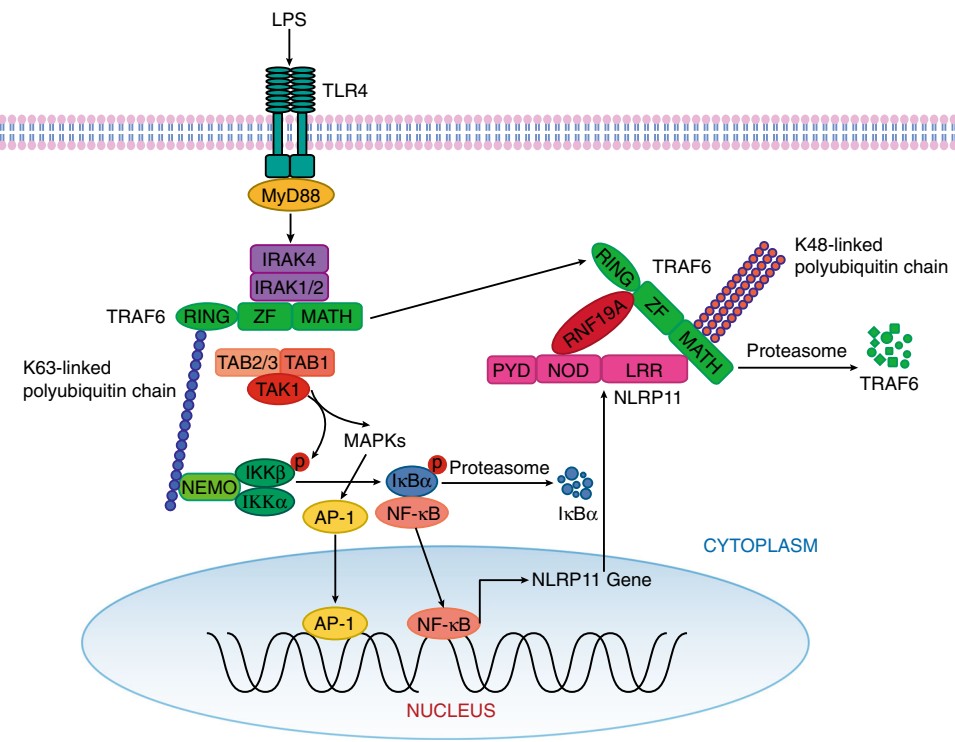

**Fig. 8** Proposed working model of NLRP11 NLRP11 expression is under the control of TLR-induced NF-κB signalling. However, its expression negatively regulates TLR-mediated NF-κB signalling by targeting TRAF6 for degradation through interaction with RNF19A for K48-linked ubiquitination

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

# ARTICLE

35. Tian, X., Pascal, G. & Monget, P. Evolution and functional divergence of NLRP genes in mammalian reproductive systems. *BMC Evol. Biol.* **9**, 202 (2009).

36. McDaniel, P. & Wu, X. Identification of oocyte-selective NLRP genes in rhesus macaque monkeys (Macaca mulatta). *Mol. Reprod. Dev.* **76**, 151–159 (2009).

37. Khare, S. et al. An NLRP7-containing inflammasome mediates recognition of microbial lipopeptides in human macrophages. *Immunity* **36**, 464–476 (2012).

38. Benko, S., Magalhaes, J. G., Philpott, D. J. & Girardin, S. E. NLRC5 limits the activation of inflammatory pathways. *J. Immunol.* **185**, 1681–1691 (2010).

39. Meissner, T. B. et al. NLR family member NLRC5 is a transcriptional regulator of MHC class I genes. *Proc. Natl Acad. Sci. USA* **107**, 13794–13799 (2010).

40. Allen, I. C. et al. NLRP12 suppresses colon inflammation and tumorigenesis through the negative regulation of noncanonical NF-kappaB signalling. *Immunity* **36**, 742–754 (2012).

41. Bhoj, V. G. & Chen, Z. J. Ubiquitylation in innate and adaptive immunity. *Nature* **458**, 430–437 (2009).

42. Jiang, X. & Chen, Z. J. The role of ubiquitylation in immune defence and pathogen evasion. *Nat. Rev. Immunol.* **12**, 35–48 (2012).

43. Heaton, S. M., Borg, N. A. & Dixit, V. M. Ubiquitin in the activation and attenuation of innate antiviral immunity. *J. Exp. Med.* **213**, 1–13 (2016).

44. Guo, Z. L. et al. Genetically modified "obligate" anaerobic Salmonella typhimurium as a therapeutic strategy for neuroblastoma. *J. Hematol.Oncol.* **8**, 99 (2015).

45. Cui, J. et al. NLRP4 negatively regulates type I interferon signalling by targeting the kinase TBK1 for degradation via the ubiquitin ligase DTX4. *Nat. Immunol.* **13**, 387–395 (2012).

46. Abe, T. et al. Germ-Cell-Specific inflammasome component NLRP14 negatively regulates cytosolic nucleic acid sensing to promote fertilization. *Immunity* **46**, 621–634 (2017).

47. Shi, H. et al. NLRP3 activation and mitosis are mutually exclusive events coordinated by NEK7, a new inflammasome component. *Nat. Immunol.* **17**, 250–258 (2016).

48. He, Y., Zeng, M. Y., Yang, D., Motro, B. & Nunez, G. NEK7 is an essential mediator of NLRP3 activation downstream of potassium efflux. *Nature* **530**, 354–357 (2016).

## Acknowledgements

This work was supported by the National Natural Science Foundation of China (31370869, 91629101, 31522018, and 81572766), the National Key Basic Research Program of China (2015CB859800, 2014CB910800, 2014CB745203, and 2017YFA0103802), Shenzhen Peacock Plan (KQTD20130416114522736), Shenzhen Technology Research Program (JSGG20160301161836370), Guangdong Natural Science Funds for Distinguished Young Scholar (S2013050014772), Guangdong Innovative Research Team Program (2011Y035 and 2016ZT06S029), the Science and Technology Planning Project of Guangdong Province (2015B020228002), and was in part supported by grants (CA101795) from NCI, NIH (to R.-F.W.).

## Author contributions

Conceptualization, J.C. and R.-F.W.; methodology, C.W., Z.S., M.L., J.O., and J.C.; experiments, C.W., Z.S., M.L., and J.O.; writing, original draft, C.W., R.-FW., and J.C.; writing, review and editing, R.-F.W. and J.C.; funding acquisition, W.Z., R.-F.W. and J.C.; supervision, R.-F.W., and J.C.

## Additional information

**Competing interests:** The authors declare no competing financial interests.

