## [Peer Review File · Nature Communications]

Reviewers' comments:

Reviewer #1 (Remarks to the Author):

Wu et al, present a very interesting, well-written manuscript demonstrating the role of NLRP11 in human myeloid cells as a negative regulator of TLRs using strong and thoughtful biochemical and cell biological approaches. The authors elegantly demonstrate NLRP11 recruits the E3 Ub ligase RNF19A to mediate the degradation of TRAF6 and thus limit TLR induced NF κ B responses. The role NLRP11 was previously unknown and as such this well conceived and thorough manuscript is of great interest and significance to the innate immune field, and some interest to the wider scientific community.

I have a few minor concerns to be addressed:

Fig1:

1. Can other NF κ B inducers (non-TLR) induce NLRP11 expression e.g. TNF, PMA?
2. Please clarify if THP-1 cells are PMA differentiated or not.
3. What is the expression in primary human monocytes vs PBMCs? Is the expression in PBMCs mostly coming from monocytes?
4. Is NLRP11 expressed in other cell types upon TLR activation? e.g. DCs?

S.Fig1c:

5. DAPI on figure misspelled as DIPA

Fig 2:

6. In 2b; it would be nice to also show P-IRF3 or TBK-1 as a control, which should remain unaltered.
7. In 2e; this figure is redundant with 2f for LPS and could be removed. In Fig2f it would be good to include polyI:C as a control that should remain unaltered by NLRP11 overexpression.

Fig3:

8. Performing an in vitro bacterial infection model in WT vs NLRP11 KO THP-1 cells to add more physiological relevance of the findings/significance of NLRP11 function.

General comments:

9. For all densitometry (e.g. Fig2c, 5b etc.): unless its combined from at least three individual experiments, i.e. error bars and maybe stats, it's not informative and could be removed.
10. It is still unclear to me if much of the data is combined from 3 expts or representative of 3 expts? If combined data is shown, mean \pm SEM would be more appropriate than SD and state "combined" in legends and methods. If the data is representative of 3 expts and error bars represent technical differences, stats should not be presented and state "representative data" in legends and methods.

Reviewer #2 (Remarks to the Author):

Wu et al report in their paper "NLRP11 attenuates Toll-like receptor signaling by targeting TRAF6 for degradation via the ubiquitin ligase RNF19A" a number of well-designed experiments to prove that TRAF6 is a specific target for the the NLP11 -RNF19A complex, causing degradation of TRAF6. They provide a number of data for their hypothesis using siRNA KO and CRISPR-Cas9 genome editing, as well as, overexpression and mutagenesis, to identify acceptor lysines in TRAF6. I am convinced by the data presented by the authors, but I have few comments on the Figures.

Specific comments:

Figure 1a; pancreas is misspelled, please correct.

In Figure Legend for Figure 1;

a, Please include explanations for what kind of cells THP cells are,

b, explain for the reader what Pam3CSK4 and Bay11-7082 are used for.

Figure 1i and j; label on y-axis is missing.

Figure 1 k; Are the constructs used Flag-tagged? In Figure legend it says HA-NLRP11, please check and correct.

Reviewer #3 (Remarks to the Author):

In the manuscript entitled „NLRP11 attenuates Toll-like receptor signalling by targeting TRAF6 for degradation via the ubiquitin ligase RNF19A“, Wu and co-workers report on a novel function of the human NLR protein NLRP11 in TLR responses.

NLRP11 is a primate specific protein that has been associated with oocyte development. However, alternative roles and the molecular functions of NLRP11 remain largely elusive at present. The author provide an expression analysis of NLRP11 and show that in human cell lines, myeloid THP1 cells and primary PBMCs NLRP11 affects TLR-mediated NF- κ B activation. Using classical biochemical analysis and the use of knock-down but also CRISPR knock-out lines, the authors show that NLRP11 negatively regulates TLR-induced pro-inflammatory responses by targeting TRAF6 for K48 ubiquitination-mediated proteasomal degradation. Finally, they identify the E3 ligase to be essential for this process. Their data suggests that NLRP11 recruits RNF19A to TRAF6 upon TLR activation to target TRAF6 for proteasome degradation.

Overall, the manuscript is very well written and logically organized. The provided data seems to be of good quality. The work would add to our understanding of the function of NLRP11 and, to the knowledge of this reviewer, is the first report analysing the molecular function of human NLRP11. Unfortunately, in the current form the manuscript suffers from several shortcomings that need to be addressed. The major criticisms of this reviewer being the sloppy statistics, lack of information on the robustness of the experiments, lack of specificity controls and the insufficient validation of the authors conclusions in primary cells.

Major points:

The reader could get the wrong impression that NLRP11 and RNF19A are conserved regulators of inflammatory responses. *Nlrp11* however is a primate specific gene. The fact that NLRP11 and eventually RNF19A only have primate specific functions needs to be clearly stated in the abstract and main text and a more throughout discussion of this point needs to be provided. It also would be helpful for the reader if the authors could share their views on this topic. The immunoblot shown in Fig. S3 is no proof of lack of NLRP11 in mouse as the specificity of the antibody is not provided. The authors need to expand on this statement in line 92 (eventually the authors want to refer here to a nice paper on this subject by Tian et. 2009).

This raises the important and exiting question how TRAF6 is regulated in mice. The authors might have ideas on this as they discuss that TRIM38 might function in murine but not human macrophages. It would strengthen the work if the authors could provide experimental data on this.

In view of the above, the author need to provide further evidence to substantiate the claimed general function of the proposed role of NLRP11 or show that this is specific for humans. To this end, they could analyse NLRP11 function in cells from other primates.

The work in its current form heavily relies on overexpression and the use of HEK293T cells. This is suited to gain molecular insights, however the key results need to be confirmed in primary cells.

More experimental data using primary cells is mandatory, in particular for the function of RNF19A which was only analysed in HEK293 cells. Here additional experiments using THP1 and PBMC are needed to substantiate these data.

In general, information on repetitions and statistical information is sparse and appears to be sloppy. For the supplementary figures and fig. 1h,g,l , fig.2c among others, any statistical information is missing. This impairs interpretation of the data. Descriptive statistics (S.D.) needs to be presented for all quantitative measurements and these need to be based on at least three independent replications for statistical analysis.

Apparently, there are errors in the statistical analysis, as exemplified by the experiment shown in fig. 1: According to the description, this is based on three independent experiments and presents the S.D. and mean (see MM part, statistics). However, this obviously is not correct (no SD shown). In the view of this reviewer, error bars are also not in line with the S.D. one typically would obtain in such experiments (see for example Fig. 3 h), but rather look like SEM. The authors are requested to carefully revise the statistics and perform additional experiments and reanalyse the data in cases where data seems to represent only one single experiment.

All densitometry measurements need are based on at least three independent experiments and should include descriptive statistics (mean + SD). In the figures, the signals for the loading controls appear to be saturated. The authors need to assure that the technique used guarantees that the signals were recorded and processed in a linear manner.

The screening data shown in fig. S3 are of insufficient quality in the presented form. The experiments needs to be repeated at least three times and S.D. needs to be provided. The material methods part, including the sequences used for this part of the work are missing. Work using PBMC isolation should include material from three different donors. To the experience of this reviewer, data and S.D. values as shown in Fig. 1c can unlikely be obtained for such experiments.

The manuscript lacks evidence for the claimed specificity of NLRP11 for TRAF6-mediated NF-kB responses. Suitable experiments should be added to be able to draw this conclusion. One simple experiment could be the overexpression of TBK1 and analysis of IFN responses in HEK cells in line with the experiments shown in Fig.4. Additionally, a more thoroughly analysis of the NLRP11 ko THP1 cells for their responses to other MAMP and other innate immune pathways should be provided to strengthen the manuscript and provide evidence for the claimed specific function of NLRP11 in TRAF6-mediated responses.

The author's data suggest that three lysine residues are responsible for K48-ubiquitination of TRAF6. This data should be substantiated by MS analysis to exclude that the 3KR mutant used for these experiments might be affected in folding capacity or stability.

Fig.1g: THP1 cells do not well respond to poly (I:C). The authors need to control activation of the cells by poly(I:C) to substantiate their conclusions.

Fig.1k: Could the authors provide sub cellular fractionation analysis or IF for endogenous NLRP11?

Fig.1J: It would add additional information if a kinetic according that shown in panel I could be provided.

Fig.3a,b: Probing for NLRP11 to validate knock-down in the experiment is missing. It would be nice if the authors could also add densitometric analysis of this data from the three experiments.

Fig.4e: This blot is nor very convincing as there is quite some background in the control IP for NLRP11. The authors should optimize the protocol or show a more representative blot.

Fig.4g: Strangely, there is less TRFA6 in the 1h and 2h IP, albeit the input signal is the same.

Might there be a labelling mistake and the picture present anti-NLRP11 IP?

Fig.5: Some panels include redundant data. This reviewer suggest to combine the panels a,f and g in one experiment for more clarity.

Fig.7b: Probing for RNF19A, DTX4 and RNF7 to validate the knock-down efficiency in this experiment is missing.

In databases, a high expression of NLRP11 in testis has been reported. This is in discrepancy to the expression data shown in Fig.1a. The authors are requested to validate that they see no expression of NLRP11 in testis and discuss this point.

Minor points:

Line 89: Not all NLR have a PYD, this sentence needs to be rephrased.

Line 93: Expression of NLRP11 in spleen is not reported in Ref. 35. This should be corrected.

Line 458: 10 μ M : should this read 10 nM ?

The paper by Khare et al. cited by the authors showed that heat killed *Acholeplasma laidlawii* lysates did not induce NLRP11 expression nor did NLRP11 affect IL-1beta responses. This finding is in contrast with the author's data and needs to be discussed.

The view of the authors on the oocyte specific expression of NLRP11 and the link of their data to this fact should be discussed.

Fig.1a: Spelling mistakes in legend.

Response to the comments of Reviewer #1

minor concerns

(1) Can other NF- κ B inducers (non-TLR) induce NLRP11 expression e.g. TNF, PMA?

Response: To address this question, we detected the expression of NLRP11 in THP-1 cells in response to TNF α or PMA stimulation and found that NLRP11 was induced by both TNF α and PMA treatment (**New Figure 1**), suggesting that NLRP11 is a common NF- κ B inducible gene.

New Figure 1. Real-time PCR analysis of NLRP11 expression in THP-1 cells in response to TNF α or PMA treatment.

(2) Please clarify if THP-1 cells are PMA differentiated or not.

Response: Thanks for the reviewer's comment, we have now labeled THP-1 cells as

PMA differentiated (THP-1 derived macrophages) (when appropriate) in the manuscript and Figure legends.

(3) What is the expression in primary human monocytes vs PBMCs? Is the expression in PBMCs mostly coming from monocytes?

Response: We isolated human T cells, B cells, and monocytes from PBMCs, and examined the expression of NLRP11 in these primary cells (**New Figure 2, related to Figure 1b in the manuscript**). We observed a high expression of NLRP11 in monocytes, and a modest expression of NLRP11 in T cells.

New Figure 2. Real-time PCR analysis of NLRP11 expression in PBMCs, T cells, B cells, and monocytes.

(4) Is NLRP11 expressed in other cell types upon TLR activation? e.g. DCs?

Response: We examined the expression of NLRP11 in DCs and PBMCs and found that the expression of NLRP11 in DCs is very low, compared to that in the PBMCs (**New Figure 3**). Although NLRP11 can be up-regulated in PBMCs by LPS or Pam3CSK4 treatment, it could not be induced by these stimuli in DCs (**New Figure 3**), indicating that NLRP11 does not express or can be induced upon TLR activation in DCs.

**New Figure 3.** Real-time PCR analysis of NLRP11 expression in PBMCs or DCs in response to LPS or Pam3CSK4.

(5) DAPI on figure misspelled as DIPA

**Response:** We have corrected this misspelling in supplementary figure 1b.

(6) In 2b, it would be nice to also show P-IRF3 or TBK-1 as a control, which should remain unaltered.

**Response:** As suggested by the reviewer, we have examined the phosphorylation of TBK1 in figure 2b, and found that overexpression of NLRP11 did not affect the phosphorylation of TBK1 (**New Figure 4, related to Figure 2b in the manuscript**).

New Figure 4. THP-1^{EV} and THP-1^{NLRP11} cells were stimulated with LPS (100 ng/ml) for the indicated periods, then analyzed by immunoblot with indicated antibodies.

(7) In 2e; this figure is redundant with 2f for LPS and could be removed. In Fig2f it would be good to include polyI:C as a control that should remain unaltered by NLRP11 overexpression.

Response: Thanks for the reviewer's suggestion. We have removed figure 2e and included poly(I:C) as a control in figure 2f (**New Figure 5, related to Figure 2e in the manuscript**).

New Figure 5. THP-1^{EV} and THP-1^{NLRP11} cells were stimulated with Pam3CSK4 (100 ng/ml), poly(I:C) (10 μ g/ml), LPS (100 ng/ml), or CL097 (1 μ g/ml) for 24 h, then the supernatants were analyzed by enzyme-linked immunosorbent assays (ELISAs).

(8) Performing an in vitro bacterial infection model in WT vs NLRP11 KO THP-1 cells to add more physiological relevance of the findings/significance of NLRP11 function.

Response: We appreciate the reviewer for this important suggestion. To address this question, we analyzed the activation of NF- κ B and MAPKs signaling in WT and NLRP11 KO THP-1 cells after exposure to Heat Killed Salmonella typhimurium (HKST, a TLR2, TLR4, and TLR5 agonist). We found that the activation of NF- κ B and MAPKs signaling was greatly enhanced in NLRP11 deficient cells compared to WT THP-1 cells (**New Figure 6, related to Figure 3g in the manuscript**). Consistently, the production of pro-inflammatory cytokines such as TNF- α and IL-6 was significantly increased in NLRP11-deficient cells than in WT THP-1 cells after

infected with HKST (New Figure 7, related to Figure 3h in the manuscript).

New Figure 6. WT and *NLRP11* KO THP-1 cells were infected with HKST with indicated periods, and the whole cell lysates were subjected to immunoblot analyses with indicated antibodies.

New Figure 7. WT or *NLRP11* KO THP-1 cells were exposed to HKST for 24 h and the production of TNF α and IL-6 was measured by ELISAs.

General comments:

(9) For all densitometry (e.g. Fig2c, 5b etc.): unless its combined from at least three individual experiments, i.e. error bars and maybe stats, it's not informative and could be removed.

Response: We reanalyzed the densitometry of Fig. 2c and removed the densitometry analysis of Fig. 5b as the reviewer suggested. The corresponding three independent

experimental figures of Fig. 2c are shown in **New Figure 8** and the graphs of densitometry analysis are shown in **New Figure 9** (**Figure 8** and **Figure 9**, related to **Supplementary Figure 2b** and **Figure 2c** in the manuscript, respectively).

New Figure 8. THP-1^{EV} and THP-1^{NLRP11} cells were stimulated with LPS (100 ng/ml) for the indicated periods, then the cell lysates were analyzed by immunoblot with indicated antibodies.

New Figure 9. The intensities of the indicated bands in New Figure 8 were quantified, and the ratios of intensities of the corresponding bands were calculated and are shown

in the graphs as means \pm SD from three independent experiments.

(10) It is still unclear to me if much of the data is combined from 3 expts or representative of 3 expts? If combined data is shown, mean \pm -SEM would be more appropriate than SD and state “combined” in legends and methods. If the data is representative of 3 expts and error bars represent technical differences, stats should not be presented and state “representative data” in legends and methods.

Response: We have added the statistical information of these data including Figure 1a, c, h, i, j, Figure 2c, Figure 3h and Supplementary Figure 3b in the previous manuscript (**Figure 1a, c, h, i, j, Figure 2c, Figure 3h and Supplementary Figure 3b, related to Figure 1a, d, i, j, k, Figure 2c, Figure 3f and Supplementary Figure 5d in the revised manuscript, respectively**). All these data are based on at least three independent experiments and S.D. are provided.

Response to the comments of Reviewer #2

Specific comments:

(11) Figure 1a; pancreas is misspelled, please correct.

Response: Thanks for the reviewer’s suggestion. We have now corrected this misspelling in Figure 1a.

(12) In Figure Legend for Figure 1;

a, Please include explanations for what kind of cells THP cells are,

b, explain for the reader what Pam3CSK4 and Bay11-7082 are used for.

Response:

a, We have added that THP-1 cells are one kind of human monocytic leukemia cells used in the manuscript when we firstly mentioned it in Figure 1.

b, We have added for the reader that Pam3CSK4 is a TLR1/2 ligand and Bay11-7082 is a NF- κ B inhibitor used in the manuscript and figure 1 legend.

(13) Figure 1i and j; label on y-axis is missing.

Response: We have added the missing y-axis in Figure 1i and j.

(14) Figure 1 k; Are the constructs used Flag-tagged? In Figure legend it says HA-NLRP11, please check and correct.

Response: We have corrected this error.

Response to the comments of Reviewer #3

Major points:

(15) The reader could get the wrong impression that NLRP11 and RNF19A are conserved regulators of inflammatory responses. Nlrp11 however is a primate specific gene. The fact that NLRP11 and eventually RNF19A only have primate specific functions needs to be clearly stated in the abstract and main text and a more throughout discussion of this point needs to be provided. It also would be helpful for the reader if the authors could share their views on this topic.

Response: We appreciate the reviewer's remarks and useful suggestions. NLRP11 and RNF19A are not conserved regulators of inflammatory response. We stated and discussed that NLRP11 and RNF19A specifically function in primate cells in the abstract and on page 9 of the revised manuscript, as the reviewer suggested.

(16) The immunoblot shown in Fig. S3 is no proof of lack of NLRP11 in mice, as the specificity of the antibody is not provided. The authors need to expand on this statement in line 92 (eventually the authors want to refer here to a nice paper on this subject by Tian et. 2009).

Response: As suggested by the reviewer, we have cited the paper by Tian et.2009 as the reference 35 as the reviewer suggested in the description of Figure 1.

(17) This raises the important and exiting question how TRAF6 is regulated in mice. The authors might have ideas on this as they discuss that TRIM38 might function in murine but not human macrophages. It would strengthen the work if the authors could provide experimental data on this.

Response: A previous study has confirmed that TRIM38 inhibited TLRs signaling by targeting TRAF6 for degradation in mouse cells¹. However, we found that knockdown of TRIM38 in human derived-macrophages had no effect on the expression of TRAF6 (**New Figure 10, related to Supplementary Figure 5b in the Supplementary Materials**). In addition, TRIM38 did not promote NLRP11-mediated degradation of TRAF6 degradation (**Supplementary Figure 5c**). In this study, we demonstrated that NLRP11 could recruit another E3 ligase RNF19A to degrade TRAF6. Taken together, these results suggested that human and mice may harness different mechanisms to degrade TRAF6 to tightly regulate the activation of TLR signaling pathway. We have discussed it on page 15 and 19 in the manuscript.

New Figure 10. Immunoblot analysis of extracts of THP-1 derived-macrophages transfected with *TRIM38* siRNA or control siRNA and simulated with LPS for indicated periods. Bottom, RT-PCR analysis of *TRIM38* mRNA.

(18) In view of the above, the author need to provide further evidence to substantiate the claimed general function of the proposed role of NLRP11 or show that this is specific for humans. To this end, they could analyse NLRP11 function in cells from other primates.

Response: We appreciated the reviewer's suggestion and analyzed NLRP11 function in Rhesus macaca monkey cells. We found that knockdown of NLRP11 in Rhesus macaca monkey PBMCs increased the expression of proinflammatory cytokines as in human PBMCs (**New Figure 11, related to Supplementary Figure 3i in the Supplementary Materials**), suggesting that the biological function of NLRP11 may be conserved in primates.

New Figure 11. Rhesus macaca monkey PBMCs that were transfected with NLRP11 siRNA or control siRNA for 48 h were stimulated with LPS (100 ng/ml) for indicated periods, then analyzed by real-time PCR for *TNFA*, *IL6* and *IL1B* transcript.

(19) The work in its current form heavily relies on overexpression and the use of HEK293T cells. This is suited to gain molecular insights, however the key results need to be confirmed in primary cells. More experimental data using primary cells is mandatory, in particular for the function of RNF19A which was only analysed in HEK293 cells. Here additional experiments using THP1 and PBMC are needed to substantiate these data.

Response: We appreciated the reviewer's suggestions and performed many new experiments in THP-1 and PBMCs to confirm our hypothesis (**See New Figure 6, 7, 11, 12, 13, 17, 18**). Particularly, we analyzed the function of RNF19A in THP-1 cells and PBMCs. As shown in **New Figure 12 (related to Figure 7b in the manuscript)**,

knockdown of NLRP11 in PBMCs resulted in higher expression of several proinflammatory cytokines including TNF α , IL-6, and IL-1 β in response to LPS treatment. Furthermore, we found that the production of TNF α and IL-6 was markedly increased by the knockdown of RNF19A in THP-1 cells, whereas NLRP11 failed to inhibit the production of TNF α and IL-6 in the presence of RNF19A siRNA (New Figure 13, related to Figure 7c in the manuscript). Together, these results indicate that RNF19A plays a key role in NF- κ B signaling and functions as an E3 ligase in NLRP11-mediated inhibition of TLR signaling.

New Figure 12. PBMCs that were transfected with RNF19A siRNA or control siRNA for 48 h were stimulated with LPS (100 ng/ml) for indicated periods, then analyzed by real-time PCR for *TNFA*, *IL6*, *IL1B* and *RNF19A* transcript.

New Figure 13. ELISA analysis of TNF α and IL-6 production in THP-1^{EV} and THP-1^{NLRP11} cells transfected with ctrl siRNA or RNF19A siRNA #1 and stimulated with LPS for 0 or 24 hours.

THP-1^{NLRP11} cells that were transfected with RNF19 siRNA and stimulated with LPS for 24 h.

(20) In general, information on repetitions and statistical information is sparse and appears to be sloppy. For the supplementary figures and fig. 1h,g,i, fig.2c among others, any statistical information is missing. This impairs interpretation of the data. Descriptive statistics (S.D.) needs to be presented for all quantitative measurements and these need to be based on at least three independent replications for statistical analysis. Apparently, there are errors in the statistical analysis, as exemplified by the experiment shown in fig. 1: According to the description, this is based on three independent experiments and presents the S.D. and mean (see MM part, statistics). However, this obviously is not correct (no SD shown). In the view of this reviewer, error bars are also not in line with the S.D. one typically would obtain in such experiments (see for example Fig. 3 h), but rather look like SEM. The authors are requested to carefully revise the statistics and perform additional experiments and reanalyse the data in cases where data seems to represent only one single experiment.

Response: We sincerely thank the reviewer for these valuable comments. We have added the statistical information of these data including Figure 1a, c, h, i, j, Figure 2c, Figure 3h and Supplementary Figure 3b in the previous manuscript (**Figure 1a, c, h, i, j, Figure 2c, Figure 3h and Supplementary Figure 3b, related to Figure 1a, d, i, j, k, Figure 2c, Figure 3f and Supplementary Figure 5d in the revised manuscript, respectively**). All these data are based on at least three independent experiments and S.D. are provided.

(21) All densitometry measurements need are based on at least three independent experiments and should include descriptive statistics (mean + SD). In the figures, the signals for the loading controls appear to be saturated. The authors need to assure that the technique used guarantees that the signals were recorded and processed in a linear manner.

Response: To address this question, we reanalyzed the data including Figure 1i, j, k, Figure 2c, and Figure 3a, b. All these densitometry measurements are based on at least three independent experiments and descriptive statistics (mean + SD) are included. The signals for the loading controls of Figure 1i and Figure 2b in previous manuscript appear to be saturated. We replaced the loading control figures with shorter time exposure (**Figure 1i related to Figure 1j, Figure 2b related to Figure 2b in revised manuscript**).

(22) The screening data shown in fig. S3 are of insufficient quality in the presented form. The experiments needs to be repeated at least three times and S.D. needs to be provided. The material methods part, including the sequences used for this part of the work are missing.

Response: We have performed statistic analysis of the screening experiments and the S.D. has been provided (**New Figure 14**). We have also included all the shRNA sequences of E3 ligases shown in this figure in supplemental materials (**Supplementary Table 3**). (**New Figure 14, related to Supplementary Figure 5d in the Supplementary Materials**).

New Figure 14. 293T cells were first transfected with indicated shRNA from a human RNIG domain-containing E3 ubiquitin ligase shRNA sub-library for 12 h. The cells were then transfected with HA-tagged TRAF6, Flag-tagged NLRP11 and NF-κB reporter. The cell lysates were harvested 24 h post the second transfection and

subjected to a dual-luciferase assay.

(23) Work using PBMC isolation should include material from three different donors. To the experience of this reviewer, data and S.D. values as shown in Fig. 1c can unlikely be obtained for such experiments.

Response: We have examined the expression of NLRP11 in PBMCs from three different donors and performed statistical analyses of these data. (**New Figure 15, related to Figure 1d in the manuscript**).

New Figure 15. PBMCs were stimulated with LPS (100 ng/ml) for the indicated time points, and the transcript levels of NLRP11 were analyzed by real-time PCR.

(24) The manuscript lacks evidence for the claimed specificity of NLRP11 for TRAF6-mediated NF- κ B responses. Suitable experiments should be added to be able to draw this conclusion. One simple experiment could be the overexpression of TBK1 and analysis of IFN responses in HEK cells in line with the experiments shown in Fig.4.

Additionally, a more thoroughly analysis of the NLRP11 ko THP1 cells for their responses to other MAMP and other innate immune pathways should be provided to strengthen the manuscript and provide evidence for the claimed specific function of

NLRP11 in TRAF6-mediated responses.

Response: As the reviewer suggested, we examined the IFN responses in HEK 293T cells and found that NLRP11 did not inhibit TBK1-induced IFN β activation (**New Figure 16, related to Figure 4b in the manuscript**). We also examined the production of TNF α and IL-6 in WT and NLRP11 KO THP-1 cells in response to other TLR ligands. We found that knockout of NLRP11 could enhance the production of TNF α and IL-6 in response to Pam3CSK4, LPS and CL097, but not poly(I:C) (**New Figure 17, related to Figure 3f in the manuscript**). Additionally, overexpression of NLRP11 in THP-1 cells had no effect on the activation of NF- κ B and MAPKs in response to TNF α (**New Figure 18, related to Supplementary Figure 2c in the Supplementary Materials**). Together, these results indicate that NLRP11 specifically inhibits TRAF6-mediated TLR signaling.

New Figure 16. IFN- β luciferase activity in 293T cells that were transfected with a IFN- β reporter and TBK1, together with increasing amounts of NLRP11 for 24 h. The results are presented relative to *Renilla* luciferase activity.

New Figure 17. WT or *NLRP11* KO THP-1 cells were stimulated with Pam3CSK4 (100 ng/ml), poly (I:C) (10 µg/ml), LPS (100 ng/ml), or CL097 (1 µg/ml) for 24 h before the supernatants were collected. The production of TNFα and IL-6 was measured by ELISAs.

New Figure 18. THP-1^{EV} and THP-1^{NLRP11} cells were stimulated with TNFα (10 ng/ml) for the indicated periods, then analyzed by immunoblot with indicated antibodies.

(25) The author's data suggest that three lysine residues are responsible for

K48-ubiquitination of TRAF6. This data should be substantiated by MS analysis to exclude that the 3KR mutant used for these experiments might be affected in folding capacity or stability.

Response: As the reviewer suggested, we performed mass spectrometry and failed to get the sequence signal of the 3KR region from either WT TRAF6 or its mutant. Instead, we examined the endogenous K48- and K63-linked ubiquitination of WT and 3KR mutant TRAF6 in TRAF6 KO cells. We found that NLRP11 could promote K48-linked ubiquitination of WT TRAF6, but had no effect on 3KR mutant. By contrast, NLRP11 had no effect on the K63-linked ubiquitination of both WT and 3KR mutations (**New Figure 19, related to Figure 6e in the manuscript**). In addition, mutation of 3KR did not affect the stability and function of TRAF6 (Figure 6f, g). Together these data suggest that 3KR mutant merely affects the K48 ubiquitination of TRAF6.

New Figure 19. TRAF6 KO 293T cells transfected with TRAF6 WT or 3KR mutant and Myc-NLRP11 were subjected to denaturing immunoprecipitation with an anti-Flag antibody followed by immunoblotting with the indicated antibodies. All the cells were treated with 10 μ M MG132 for 6 h before harvesting.

(26) Fig.1g: THP1 cells do not well respond to poly(I:C). The authors need to control activation of the cells by poly(I:C) to substantiate their conclusions.

Response: To address this question, we examined the expression of IL-6 to determine whether the NF- κ B signaling is well activated in response to Pam3CSK4, poly(I:C)

(10 µg/ml) or LPS (100 ng/ml) treatment. We found that Pam3CSK4 or LPS treatment could strongly induce the transcription of IL-6. Although PBMCs do not well respond to poly (I:C), poly (I:C) treatment still weakly triggered the transcription of IL-6 (**New Figure 20**). Thus, we proposed that the up-regulation of NLRP11 in response to TLR signaling may need strong activation of NF-κB signaling.

New Figure 20. Real-time PCR analysis of IL-6 expression in PBMCs in response to Pam3CSK4 (100 ng/ml), polyI:C (10 µg/ml), LPS (100 ng/ml), or CL097 (1 µg/ml) for 8 h.

(27) Fig.1k: Could the authors provide sub cellular fractionation analysis or IF for endogenous NLRP11?

Response: As suggested by the reviewer, we examined the sub-cellular localization of NLRP11 in THP-1 cells by sub cellular fractionation analysis using endogenous antibody to NLRP11. We found that NLRP11 mainly localized in the cytoplasm and the LPS treatment did not change the localization of NLRP11. Moreover, we found that leptomycin (LMB, which inhibits CrmA-mediated nuclear export) treatment did not alter the cytosolic localization of NLRP11 (**New Figure 21, related to Figure 11 in the manuscript**). Together these results suggest that NLRP11 is a cytoplasmic protein.

New Figure 21. The THP-1 cells were stimulated or unstimulated with LPS for 8 h, and cytoplasmic and nuclear extracts of THP-1 cells were treated or untreated with 50 nM LMB for 4 h were analyzed by immunoblotting with antibodies to NLRP11, tubulin (cytoplasmic fraction), and lamin (nuclear fraction).

(28) Fig.1J: It would add additional information if a kinetic according that shown in panel I could be provided.

Response: As the reviewer suggested, we reanalyzed the turnover rate of endogenous of NLRP11 in the presence of CHX (a protein synthesis inhibitor) and MG132 (a proteasomal inhibitor). We found that NLRP11 is rapidly degraded in the presence of CHX and accumulated in the presence of MG132, suggesting that NLRP11 is a short-life protein and rapidly degraded via the proteasomal pathway (**New Figure 22, related to Figure 1k**).

New Figure 22. THP-1 cells were untreated or pretreated with MG132 for 6h, then the cells were incubated with CHX for the indicated periods and were immunoblotted with indicated antibodies. Quantification of relative NLRP11 levels is shown in the right panel.

(29) Fig.3a,b: Probing for NLRP11 to validate knock-down in the experiment is missing. It would be nice if the authors could also add densitometric analysis of this data from the three experiments.

Response : We have probed for NLRP11 to validate the knock-down efficiency of NLRP11 in Figure. 3a and b (**New Figure 23, related to Figure 3a,b in the manuscript**). We also added the densitometry analysis of this data from the three experiments (**New Figure 24, related to Supplementary Figure 3d,e in the**

Supplementary Materials).

New Figure 23. (a,b) Immunoblot analysis of *NLRP11*, total and phosphorylated IKK, I κ B α , as well as MAPKs (p38, JNK, and ERK) in THP-1 cells (**a**) or PBMCs (**b**) transfected with the indicated siRNA and stimulated with LPS for indicated periods.

New Figure 24. (a,b) The intensities of the indicated bands from three independent experiments were quantified, and the ratios of intensities of the corresponding bands were calculated and shown in the graphs as means \pm SD from three independent experiments.

(30) Fig.4e: This blot is not very convincing as there is quite some background in the control IP for NLRP11. The authors should optimize the protocol or show a more representative blot.

Response: We have optimized the protocol and shown a more representative blot (**New Figure 25, related to Figure 4e in the manuscript**).

New Figure 25. THP-1 cells were stimulated with LPS for the indicated periods. The cell lysates were subjected to immunoprecipitation with an anti-TRAF6 antibody or control IgG, followed by immunoblotting with an anti-NLRP11 or anti-TRAF6 antibody.

(31) Fig.4g: Strangely, there is less TRFA6 in the 1h and 2h IP, albeit the input signal is the same. Might there be a labelling mistake and the picture present anti-NLRP11 IP?

Response: We optimized the protocol and showed a more representative blot (**New Figure 26, related to Figure 4g in the manuscript**).

New Figure 26. PBMCs were stimulated with LPS for the indicated periods. The cell lysates were subjected to immunoprecipitation with an anti-TRAF6 antibody or control IgG, followed by immunoblotting with an anti-NLRP11 or anti-TRAF6 antibody.

(32) Fig.5: Some panels include redundant data. This reviewer suggest to combine the panels a,f and g in one experiment for more clarity.

Response: We have combined the panels a, f and g in one experiment for more clarity,

as the reviewer suggested (**New Figure 27, related to Figure 5a in the manuscript;**
New Figure 28, related to Figure 5f in the manuscript).

New Figure 27. Immunoblot analysis of 293T cells transfected with HA-tagged TRAF6 or TRAF2, along with or without increasing amounts of the vector encoding Flag-tagged NLRP11.

New Figure 28. Immunoblot analysis of extracts from 293T cells transfected with HA-TRAF6 and FLAG-NLRP11 or the control vector for 24 h, then treated for 6 h with 10 μ M MG132, 10 mM 3-MA, 20 mM NH₄Cl, or DMSO.

(33) Fig.7b: Probing for RNF19A, DTX4 and RNF7 to validate the knock-down efficiency in this experiment is missing.

Response: We have included the results of the knockdown efficiency of RNF19A, DTX4 and RNF7 in Supplementary materials (**New Figure 29, related to Supplementary Figure 5e**).

New Figure 29. 293T cells were transfected with the indicated shRNA for 48 h. Knockdown efficiency was analyzed by real-time PCR.

(34) In databases, a high expression of NLRP11 in testis has been reported. This is in discrepancy to the expression data shown in Fig.1a. The authors are requested to validate that they see no expression of NLRP11 in testis and discuss this point.

Response: Following the reviewer's comments, we have reanalyzed the expression of NLRP11 in fresh human tissues. We purchased the more human total RNA of different tissues and found that NLRP11 is highly expressed in the testis, ovary, and lung, and weakly expressed in other tissues (**New Figure 30, related to Figure 1a in the manuscript**). The previous data that NLRP11 expression level is low in testis may due to the variation of the samples from different parts of testis.

New Figure 30. NLRP11 mRNA in different human tissues was analyzed by real-time PCR.

Minor points:

(35) Line 89: Not all NLR have a PYD, this sentence needs to be rephrased.

Response: We rephrased it as 'NLRP11 contains...'.
(36) Line 93: Expression of NLRP11 in spleen is not reported in Ref. 35. This should be corrected.

Response: We have corrected it.

(37) Line 458: 10 μ M : should this read 10 nM ?

Response: We have corrected it.

(38) The paper by Khare et al. cited by the authors showed that heat killed *Acholeplasma laidlawii* lysates did not induce NLRP11 expression nor did NLRP11 affect IL-1 β responses. This finding is in contrast with the author's data and needs to be discussed.

Response: We have showed that the induction of NLRP11 requires a strong NF- κ B activation (Fig 1), while HKAL is not a strong NF- κ B stimulus. So it is reasonable that HKAL does not induce the expression of NLRP11. We found NLRP11 negatively regulates the secretion of IL-1 β by LPS treatment, while Khare et al reported that NLRP11 did not affect the secretion of HKAL-induced IL-1 β . These discrepancy may be explained by the multiple regulation mechanisms of IL-1 β secretion. IL-1 β secretion can be regulated by NF- κ B signaling at transcriptional level, and by inflammasome activation at post-transcriptional level. LPS is a strong NF- κ B inducer, while HKAL is not. That's why NLRP11 negatively regulates LPS-induced IL-1 β as well as secretion by inhibiting NF- κ B activation at transcriptional level. However, HKAL is a weak NF- κ B activator compared to LPS and it mainly promotes IL-1 β secretion by activating inflammsomes, and NLRP11 does not affect this process. That's why NLRP11 does not affect IL-1 β secretion induced by HKAL.

(39) The view of the authors on the oocyte specific expression of NLRP11 and the link of their data to this fact should be discussed.

Response: Several NLRPs (like NLRP2, 4, 5, 7, 9, 11 and 14) were shown to specifically or preferentially expressed in mammalian oocyte². NLRP14 was recently identified as a germ-cell-specific inhibitor of cytosolic nucleic acid sensing to promote fertilization³, suggesting that controlling innate immune response is crucial to maintain proper immunological homeostasis in germline. In addition, the activation of NLRP3 inflammasome and mitosis are mutually exclusive events mediated by NEK7^{4, 5}. Here, we demonstrated that NLRP11 was highly expressed in testis and ovary and paly pivotal roles in attenuating TLR signaling, suggesting that NLRP11 may also have special functions between innate immune response and fertilization development. We have discussed it the Discussion part on page 20 of the revised

manuscript.

(40) Fig.1a: Spelling mistakes in legend.

Response: We have corrected it.

Reference:

1. Zhao, W., Wang, L., Zhang, M., Yuan, C. & Gao, C. E3 ubiquitin ligase tripartite motif 38 negatively regulates TLR-mediated immune responses by proteasomal degradation of TNF receptor-associated factor 6 in macrophages. *J Immunol* **188**, 2567-74 (2012).
2. Tian, X., Pascal, G. & Monget, P. Evolution and functional divergence of NLRP genes in mammalian reproductive systems. *BMC Evol Biol* **9**, 202 (2009).
3. Abe, T. et al. Germ-Cell-Specific Inflammasome Component NLRP14 Negatively Regulates Cytosolic Nucleic Acid Sensing to Promote Fertilization. *Immunity* **46**, 621-634 (2017).
4. Shi, H. et al. NLRP3 activation and mitosis are mutually exclusive events coordinated by NEK7, a new inflammasome component. *Nat Immunol* **17**, 250-8 (2016).
5. He, Y., Zeng, M.Y., Yang, D., Motro, B. & Nunez, G. NEK7 is an essential mediator of NLRP3 activation downstream of potassium efflux. *Nature* **530**, 354-7 (2016).

Reviewers' comments:

Reviewer #1 (Remarks to the Author):

The authors have made a valiant attempt to address the comments by the reviewers. In general I believe the authors have done a very good job and I thank them for their hard work.

However, I have a few points that must be addressed before this paper should be accepted for publication:

Major points:

1. The major issue that was still not clearly addressed is the explanation of the data presented in graphs. Please address this appropriately.

The authors have stated "All these data are based on at least three independent experiments and SD are provided". That remains non-descriptive. This must be addressed clearly or the data (which appears sound) becomes very hard to find convincing.

I previously asked if data could be described as either "combined from" or "representative of" x number of individual experiments. This was not done, but should be upon this revision.

Where data is combined, then SEM is the appropriate way to present the error bars. Statistical analysis can be applied to these data.

If data is representative then error bars must be explained (e.g. error bars represent three technical replicates) and SD must be used. Statistical analyses should not be performed on representative data as the error is simply due to technical variance not biological variance.

2. The authors must add how THP-1-derived macrophages are generated in the methods. How much PMA? How long? etc... This was not addressed appropriately in the revised manuscript.

Minor points:

3. Why are the new data from the rebuttal New Fig1 and New Fig 3 not included in the manuscript? Please consider adding to the main or supplementary figures. They add value to the manuscript.

4. Please state that Bay-11 is in fact an inhibitor of I κ B α , which therefore indirectly inhibits NF κ B.

Once these points are suitably addressed I believe this manuscript should be accepted for publication.

Reviewer #3 (Remarks to the Author):

The authors address all my queries. They provide novel experimental data that significantly strengthens their conclusions.

However, the newly provided data for the expression analysis of Nlrp11 shown in Fig. 1 panel a and b are unclear for this reviewer.

The data shown in Fig.1a is now opposite the initial provided data set. Moreover, the data shown for expression of Nlrp11 in T cells, B cells and monocytes does not match reported expression profiles of Nlrp11 in databases (see for example <http://biogps.org/#goto=genereport&id=20480>). To clarify the nature of this relevant discrepancy, the author should conduct measurement of yet

an other independent cDNA panel and comment in the text on the discrepancies with their expression data and database sets.

Response to the comments of Reviewer #1

Major points:

(1) The major issue that was still not clearly addressed is the explanation of the data presented in graphs. Please address this appropriately. The authors have stated “All these data are based on at least three independent experiments and SD are provided”. That remains non-descriptive. This must be addressed clearly or the data (which appears sound) becomes very hard to find convincing. I previously asked if data could be described as either “combined from’ or “representative of” x number of individual experiments. This was not done, but should be upon this revision.

Where data is combined, then SEM is the appropriate way to present the error bars. Statistical analysis can be applied to these data. If data is representative then error bars must be explained (e.g. error bars represent three technical replicates) and SD must be used. Statistical analyses should not be performed on representative data as the error is simply due to technical variance not biological variance.

Response: To address this comments, we have now clarified how data were generated and analyzed as following:

1) The data generated by real-time PCR assay (**Figure 1a-d, i, Figure 2d, Figure 5c, Figure 7b, Supplementary Figure 1b-d, Supplementary Figure 2a, Supplementary Figure 3a, f, h, i, and Supplementary Figure 5a, e, f**), ELISA assay (**Figure 2e, Figure 3f, h, Figure 7c, and Supplementary Figure 3g**), and the

luciferase assay (**Figure 4a, b, j, Figure 6g, Figure 7a up panel, and Supplementary Figure 5 d, g, h**) are combined from at least three independent experiments performed in triplicate and SEM are provided.

2) The data generated by densitometric analyses (**Figure 1j right panel, k right panel, Figure 2c, and Supplementary Figure 3 d, e**) are combined from three individual experiments and represent means \pm SEM.

3) The immunoblot results (**Figure 1 e-h, j left panel, k left panel, l, Figure 2 a, b, Figure 3 a, b, d, e, g, Figure 4 d-g, i, Figure 5 a, b, d-f, h, i, Figure 6a, c-f, Figure 7 a down panel, d, f-k, Supplementary Figure 1 a, f, Supplementary Figure 2 b, c, Supplementary Figure 3 b, c, Supplementary Figure 4 b, c, and Supplementary Figure 5 b, c**) are representatives of at least three independent experiments.

We have included this information in the figure legends for clarity in the revised manuscript.

(2) The authors must add how THP-1-derived macrophages are generated in the methods. How much PMA? How long? etc... This was not addressed appropriately in the revised manuscript.

Response: THP-1 cells were seeded in 6-well plate at the density of 1×10^6 cells/ml. THP-1 cells were differentiated to macrophages by stimulation with 60 nM PMA for 16 h, and then cultured for an additional 48 h in fresh medium prior to further treatment. This experimental method has been added into the Methods section on page 22 in the manuscript.

Minor points:

(3) Why are the new data from the rebuttal New Fig1 and New Fig 3 not included in the manuscript? Please consider adding to the main or supplementary figures. They add value to the manuscript.

Response: Thanks for the reviewer's comment. We have already included these data in Supplementary Figure 1d and c. (**New Figure 1 and New Figure 3 in the last response letter, related to Supplementary Figure 1d and 1c, respectively**).

(4) Please state that Bay-11 is in fact an inhibitor of I κ B α , which therefore indirectly inhibits NF κ B.

Response: We have stated that Bay-11 is an inhibitor of I κ B α , which indirectly inhibits NF- κ B on page 6 and 32 of the manuscript.

Response to the comments of Reviewer #2

(5) However, the newly provided data for the expression analysis of Nlrp11 shown in Fig. 1 panel a and b are unclear for this reviewer.

The data shown in Fig.1a is now opposite the initial provided data set.

Response: Thanks for raising this important point. Reviewer #3 found that a high expression of NLRP11 in testis has been reported before, which is contradicted with the result of Figure 1a in our original submission. Previous RNA samples used might have some degradation. To address this question, we purchased a new set of RNA samples of human tissues, and found that NLRP11 is highly expressed in testis, ovary, and weakly expressed in other tissues. We, therefore, believe that new data were more reliable correct.

(6) Moreover, the data shown for expression of Nlrp11 in T cells, B cells and monocytes does not match reported expression profiles of Nlrp11 in databases (see

for example <http://biogps.org/#goto=genereport&id=20480>). To clarify the nature of this relevant discrepancy, the author should conduct measurement of yet another independent cDNA panel and comment in the text on the discrepancies with their expression data and database sets.

Response: The database results are obtained from the high-throughput approaches. In many cases, these genome-scale results can be used for a general trend, but need to be validated by the experiments using real-time PCR with specific primers. To clarify this issue, we reanalyzed the expression of Nlrp11 in T cells, B cells, monocytes and PBMCs by real-time PCR using another independent cDNA source. We found that Nlrp11 was highly expressed in monocytes at mRNA level (**New Figure 1**), which is consistent with our previous data. In addition, we found that Nlrp11 was highly expressed in monocytes at protein level (**New Figure 2, related to Supplementary Figure 1a**), which is consistent with our real-time PCR result. Thus, the expression of Nlrp11 in T cells, B cells, monocytes and PBMCs is true and reasonable.

New Figure 1. Real-time PCR analysis of NLRP11 expression in PBMCs, T cells, B cells, and monocytes.

New Figure 2. Immunoblot analysis of NLRP11 in PBMCs, T cells, B cells, and

monocytes.

Thank you for the opportunity to improve the quality of our manuscript by incorporated these suggested changes.

REVIEWERS' COMMENTS:

Reviewer #1 (Remarks to the Author):

Thank you for clarifying your stats and revising your manuscript. I now believe this manuscript should be accepted for publication.

Reviewer #3 (Remarks to the Author):

The authors now accurately describe their data.
They conducted novel experiments and addressed fully my queries.